 eLife

# Divisive suppression explains high-precision firing and contrast adaptation in retinal ganglion cells

Yuwei Cui[1,2], Yanbin V Wang[3,4], Silvia J H Park[3], Jonathan B Demb[3,4]*, Daniel A Butts[1,2]*

[1]Department of Biology, University of Maryland, College Park, United States; [2]Program in Neuroscience and Cognitive Science, University of Maryland, College Park, United States; [3]Department of Ophthalmology and Visual Science, Yale University, New Haven, United States; [4]Department of Cellular and Molecular Physiology, Yale University, New Haven, United States

**Abstract** Visual processing depends on specific computations implemented by complex neural circuits. Here, we present a circuit-inspired model of retinal ganglion cell computation, targeted to explain their temporal dynamics and adaptation to contrast. To localize the sources of such processing, we used recordings at the levels of synaptic input and spiking output in the in vitro mouse retina. We found that an ON-Alpha ganglion cell's excitatory synaptic inputs were described by a divisive interaction between excitation and delayed suppression, which explained nonlinear processing that was already present in ganglion cell inputs. Ganglion cell output was further shaped by spike generation mechanisms. The full model accurately predicted spike responses with unprecedented millisecond precision, and accurately described contrast adaptation of the spike train. These results demonstrate how circuit and cell-intrinsic mechanisms interact for ganglion cell function and, more generally, illustrate the power of circuit-inspired modeling of sensory processing.

*For correspondence: jonathan.demb@yale.edu (JBD); dab@umd.edu (DAB)

**Competing interests:** The authors declare that no competing interests exist.

## Introduction

Neural computations in the retina are generated by complex circuits that drive the responses of ~30 distinct ganglion cell types (**Baden et al., 2016**; **Demb and Singer, 2015**; **Sanes and Masland, 2015**). Despite the complexity of retinal circuitry, many aspects of the responses of ganglion cells to visual stimuli can be predicted with a straightforward Linear-Nonlinear (LN) cascade model (**Shapley, 2009**). In this model, a linear receptive field filters the stimulus, and a nonlinear function shapes the output by implementing the spike threshold and response saturation (**Baccus and Meister, 2002**; **Chichilnisky, 2001**; **Kim and Rieke, 2001**). However, many aspects of ganglion cell firing deviate from LN model predictions. For example, the LN model does not capture the effects of contrast adaptation, which includes a reduced gain (i.e., filter amplitude) at high contrast (**Kim and Rieke, 2001**; **Meister and Berry, 1999**; **Shapley and Victor, 1978**). The LN model also does not predict firing at high temporal resolution (**Berry and Meister, 1998**; **Butts et al., 2016, 2007**; **Keat et al., 2001**; **Passaglia and Troy, 2004**; **Uzzell and Chichilnisky, 2004**), and yet precise firing likely represents an essential element of downstream visual processing (**Bruno and Sakmann, 2006**; **Havenith et al., 2011**; **Kelly et al., 2014**; **Wang et al., 2010a**).

To improve on the LN model, several nonlinear approaches have been proposed. The first approach describes the nonlinear function between stimulus and response as a mathematical expansion, extending from the linear receptive field (**Chichilnisky, 2001**) to 'second-order' quadratic

**eLife digest** Visual processing begins in the retina, a layer of light-sensitive tissue at the back of the eye. The retina itself is made up of three layers of excitatory neurons. The first comprises cells called photoreceptors, which absorb light and convert it into electrical signals. The photoreceptors transmit these signals to the next layer, the bipolar cells, which in turn pass them on to the final layer, the retinal ganglion cells. The latter are responsible for sending the signals on to the brain. Other cells in the retina inhibit the excitatory neurons and thereby regulate their signals.

While the basic structure of the retina has been described in detail, we know relatively little about how retinal ganglion cells represent information from visual scenes. Existing models of vision fail to explain several aspects of retinal ganglion cell activity. These include the exquisite timing of ganglion cell responses, and the fact that retinal ganglion cells adjust their responses to suit different visual conditions. In the phenomenon known as contrast adaptation, for example, ganglion cells become more sensitive during small variations in contrast (differences in color and brightness) and less sensitive during high variations in contrast.

To understand how ganglion cells process visual stimuli, Cui et al. recorded the inputs and outputs of individual ganglion cells in samples of tissue from the mouse retina. By feeding these data into a computer model, Cui et al. were able to identify the mathematical calculations that take place at each stage of the retinal circuit. The findings suggest that a key element shaping the response of ganglion cells is the interaction between two visual processing pathways at the level of the bipolar cells. The resulting model can predict the responses of ganglion cells to specific inputs from bipolar cells with millisecond precision.

Future studies should extend the model to more complex visual stimuli. The approach could also be adapted to study different types of ganglion cells in order to obtain a more complete picture of the workings of the retina.

terms, using either spike-triggered covariance (*Fairhall et al., 2006*; *Liu and Gollisch, 2015*; *Samengo and Gollisch, 2013*; *Vaingankar et al., 2012*) or maximally informative dimension analyses (*Sharpee et al., 2004*). Such expansion terms better predict the spike train, but they are difficult to interpret functionally and with respect to the underlying circuitry (*Butts et al., 2011*; *McFarland et al., 2013*). The second approach targets specific aspects of the response, such as spike-refractoriness (*Berry and Meister, 1998*; *Keat et al., 2001*; *Paninski, 2004*; *Pillow et al., 2005*), gain changes associated with contrast adaptation (*Bonin et al., 2005*; *Mante et al., 2008*; *Meister and Berry, 1999*; *Shapley and Victor, 1978*), the interplay of excitation and inhibition (*Butts et al., 2016*, *2011*), and rectification of synaptic release, associated with nonlinear spatial processing (*Freeman et al., 2015*; *Gollisch, 2013*; *Schwartz and Rieke, 2011*). However, each of these models primarily focuses on one type of nonlinear computation and does not generalize to explain a range of response properties.

Here we derive a novel nonlinear modeling framework inspired by retinal circuitry. The model is constrained by recordings at two stages of processing: excitatory synaptic input and spike output, recorded in mouse retinal ganglion cells. We focused on ON-Alpha ganglion cells because they comprise a major input to lateral geniculate nucleus (LGN) and superior colliculus, and because they could be targeted routinely, in vitro, based on their large soma size. Furthermore, their spiking response to contrast modulation is mediated predominantly by synaptic excitation (*Murphy and Rieke, 2006*). We devised a tractable model of excitatory currents that incorporates a nonlinear structure based on realistic circuit elements. In particular, we allowed for divisive suppression acting on a ganglion cell's excitatory inputs to capture the computations implemented by presynaptic inhibition (*Eggers and Lukasiewicz, 2011*; *Franke et al., 2016*) and synaptic depression (*Jarsky et al., 2011*; *Ozuysal and Baccus, 2012*) at bipolar cell terminals. Ganglion cell firing, further shaped by spike generation mechanisms, could be predicted by the model with millisecond precision. Our study establishes a unified model of nonlinear processing within ganglion cells that accurately captures both the generation of precise firing events and fast contrast adaptation. Similar circuit-inspired modeling could be applied widely in other sensory systems.

## Results

We recorded spikes from ON-Alpha ganglion cells in the in vitro mouse retina while presenting a temporally modulated (<30 Hz), 1-mm spot centered on the neuron's receptive field (*Figure 1A*, *top*). Every 10 s the contrast level switched between high and low. In high contrast, the stimulus evoked spike responses that were precisely timed from trial to trial (*Figure 1A*, *left*), consistent with previous work performed both in vitro and in vivo (*Berry and Meister, 1998*; *Butts et al., 2016*; *Butts et al., 2007*; *Passaglia and Troy, 2004*; *Reinagel and Reid, 2000*; *Uzzell and Chichilnisky, 2004*). Such precision was not clearly present at low contrast (*Figure 1A*, *right*).

We first used a linear-nonlinear (LN) cascade model (*Figure 1B*) (*Chichilnisky, 2001*; *Hunter and Korenberg, 1986*) to predict the observed responses. The 'L' (linear) step of the cascade processes the stimulus with a linear 'receptive field' **k**, whose output reflects the degree that the stimulus $s(t)$ matches **k**. The 'N' (nonlinear) step acts on the output of the receptive field, $\mathbf{k}\cdot\mathbf{s}(t)$, which is scaled by a nonlinear function that could include the effects of spike threshold and response saturation. Both the linear receptive field and the nonlinearity are fit to the data in order to better predict the firing rate. The resulting receptive field had a biphasic shape at both contrasts, representing the sensitivity of the neuron to dark-to-light transitions (*Figure 1B*). Furthermore, the filter had a smaller amplitude at high contrast, a signature of contrast adaptation (*Baccus and Meister, 2002*; *Kim and Rieke, 2001*; *Zaghloul et al., 2005*).

Despite capturing the coarse temporal features of the response, the LN model could not capture fine temporal features at high contrast (*Figure 1A*) (*Berry and Meister, 1998*; *Liu et al., 2001*). To precisely compare time scales of the observed data with model predictions, we performed 'event analysis', which divides the spike train into firing events separated by silence (*Butts et al., 2010*; *Kumbhani et al., 2007*). Based on this analysis, the LN model failed to predict either the SD of the first-spike in each event or the overall event duration in high contrast, but was largely successful in low contrast (*Figure 1C*).

To improve on the LN model prediction, we included a refractory period (RP) following each spike (*Paninski, 2004*), which has previously been suggested as a mechanism for precise firing in ganglion cells (*Berry and Meister, 1998*; *Keat et al., 2001*) (see Materials and methods). However, while the resulting LN+RP model could predict the temporal properties of the spike train at low contrast, it failed at high contrast (*Figure 1C*). Thus, spike-refractoriness alone could not explain the precision at high contrast, and consequently could not predict how the response changes from low to high contrast.

## Nonlinear processing distributed across two stages of retinal processing

Because some degree of contrast adaptation is already present in a ganglion cell's excitatory synaptic inputs (*Beaudoin et al., 2007*; *Beaudoin et al., 2008*; *Kim and Rieke, 2001*), we hypothesized that we might uncover the source of the nonlinear processing by directly modeling the synaptic input currents. We therefore made whole-cell patch clamp recordings on the same neurons we recorded spike responses from, and performed a similar LN analysis on excitatory synaptic currents (*Figure 1D*). The LN model of the currents (*Figure 1E*) – like that of the spike response – accurately predicted the observed response at low contrast, but performed relatively poorly at high contrast (*Figure 1D*). To compare the precision of the LN model to the observed data, we measured the coherence between the trial-averaged response or model prediction and the responses on individual trials (see Methods); this measure captures the consistency of the response across repeats across time scales. At low contrast, the coherence of the excitatory current matched that of the LN model prediction, whereas at high contrast the coherence of the current extended to finer time scales (i.e., higher frequencies) and hence exceeded the precision predicted by the LN model (*Figure 1F*).

Contrast adaptation was measured in the synaptic currents by comparing LN models at each contrast level (*Figure 1E*). The linear filter for the current responses had a larger amplitude (i.e., higher gain) in low contrast compared with high contrast (*Beaudoin et al., 2007*; *Beaudoin et al., 2008*; *Kim and Rieke, 2001*). This adaptation occurred rapidly after the contrast switch and showed a barely discernable slow component that has been observed in other ganglion cell types (*Figure 1— figure supplement 1*; [*Baccus and Meister, 2002*; *Manookin and Demb, 2006*]). To compare the contrast-dependent gain change in the currents and spikes, we define contrast gain as the ratio

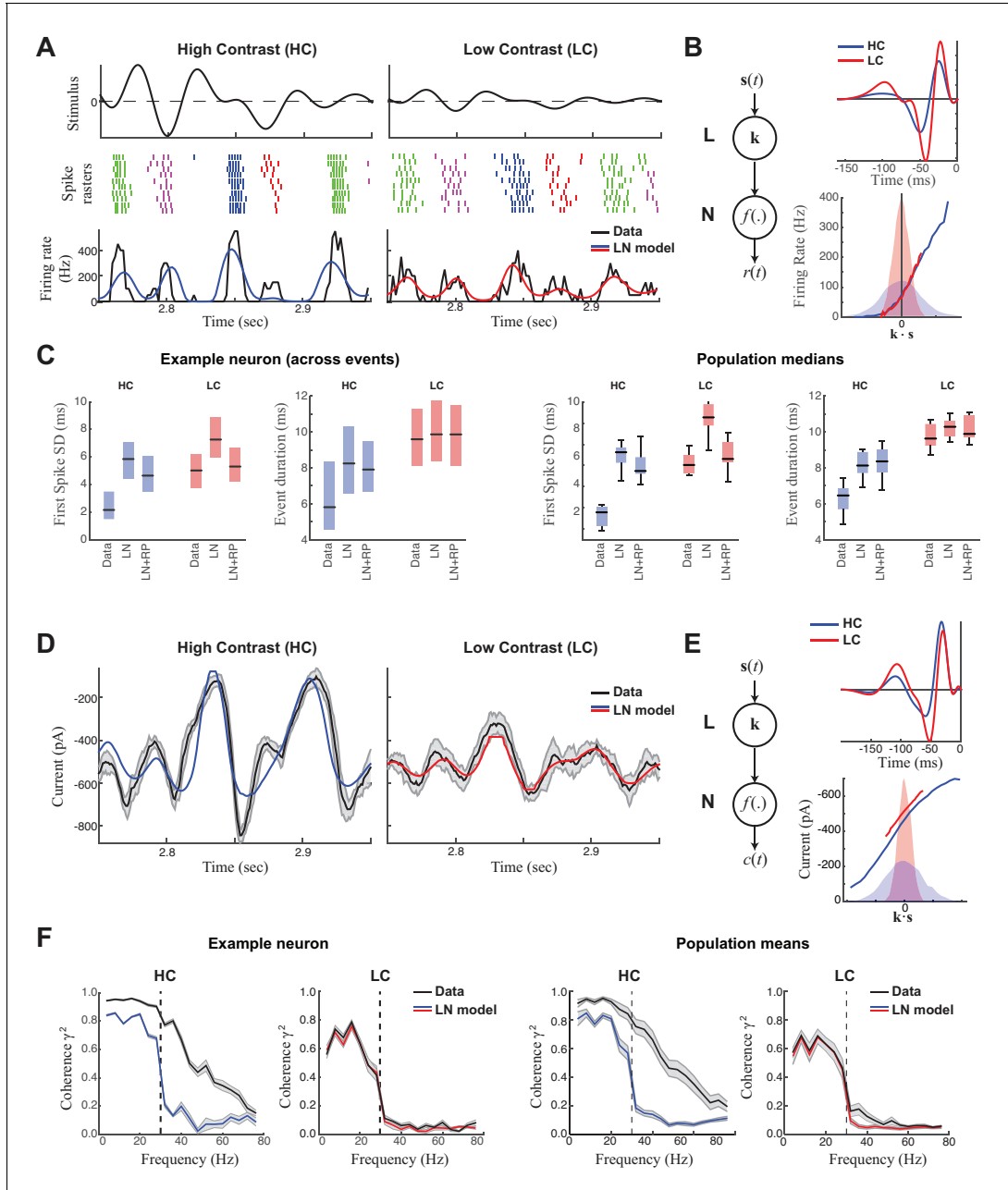

**Figure 1.** Precision of ganglion cell spike trains arises at the level of synaptic inputs. (**A**) Spike rasters of an ON-Alpha cell to 10 repeated presentations of a temporally modulated noise stimulus (*top*) at two contrast levels. The response was parsed into separate 'events' (labeled by different colors). The PSTH (*bottom*) is compared with predictions of the LN model (blue, red), which fits better at low contrast. (**B**) The LN model (schematic: *left*) was fit separately at each contrast, with the effects of adaptation isolated to the linear filters (*top*), which share the same nonlinearity (*bottom*). Nonlinearities are shown relative to the distributions of the filtered stimulus at high (shaded blue) and low (shaded red) contrasts. (**C**) Temporal properties of the observed spike trains, compared with predictions of the LN model without or with a spike-history term (LN and LN+RP). *Left*: SD of the timing of the first spike in each event. *Right*: Event duration, measured by the SD of all spikes in the event (*p<10⁻⁶, 59 events). LN and LN+RP models do not reproduce the spike precision at high contrast (HC), but the LN+RP model is adequate at low contrast (LC). (**D**) Excitatory synaptic current from the neuron in (**A**–**C**) compared with the LN model predictions. Gray area indicates SD across trials, demonstrating minimal variability. (**E**) LN model fits to the current data. The temporal filters (*top*) change less with contrast compared to spike filters (**Figure 1B**). Note here there is also a tonic offset between contrasts (**Figure 1—figure supplement 1**), captured in the vertical shift of the nonlinearity (*bottom right*). (**F**). The precision of the current response was measured using the coherence between the response on individual trials and either the observed trial-averaged response (black) or LN predictions (blue, red). Gray area shows SEM across trials (*left*) and SD across the population (*right*). The LN model fails to capture high frequency response components at HC, but agrees well with the data at LC, suggesting the precision observed in ganglion cell spike trains arises at the level of synaptic inputs.

*Figure 1 continued on next page*

*Figure 1 continued*

The following figure supplements are available for figure 1:

**Figure supplement 1.** Measurement of slow contrast adaptation.
**Figure supplement 2.** Stability of recording.

between the standard deviation of the filter in low contrast over that in high contrast. The contrast gain was significantly larger for spikes ($1.61 \pm 0.23$) than for currents ($1.10 \pm 0.14$; $p < 10^{-6}$, *unpaired two-sample t-test*, spikes: $n = 11$, current: $n = 13$) (*Zaghloul et al., 2005*); in cases where both currents and spikes were recorded in the same cell, the contrast gain was larger for spikes by $30.1\% \pm 12.0\%$ ($n = 3$). These observations suggest that both contrast adaptation and temporal precision in ON-Alpha ganglion cell spike responses are generated in large part by retinal circuitry upstream of the ganglion cell, but that further transformation occurs between currents and spikes (*Kim and Rieke, 2001*).

## The nonlinear computation underlying synaptic inputs to ganglion cells

In constructing a nonlinear description of the computation present in excitatory synaptic currents, we sought to emulate elements of the retinal circuit that shape these currents (*Figure 2A*). Excitatory synaptic inputs to ganglion cells come from bipolar cells, and bipolar cell voltage responses to our stimuli are well described by an LN model (*Baccus and Meister, 2002*; *Rieke, 2001*). This suggests that mechanisms responsible for the nonlinear behavior of the postsynaptic excitatory current are localized to the bipolar-ganglion cell synapses. Possible sources of such nonlinear behavior include presynaptic inhibition from amacrine cells, which can directly gate glutamate release from bipolar cell terminals (*Eggers and Lukasiewicz, 2011*; *Euler et al., 2014*; *Franke et al., 2016*; *Schubert et al., 2008*; *Zaghloul et al., 2007*), and synaptic depression at bipolar terminals caused by vesicle depletion (*Jarsky et al., 2011*; *Markram et al., 1998*; *Ozuysal and Baccus, 2012*).

To capture the computations that could be performed by such suppressive mechanisms, we constructed a 'divisive suppression' (DivS) model (*Figure 2A*, *bottom left*). Terms simulating bipolar cell excitation and suppression are each described by a separate LN model, with a multiplicative interaction between their outputs such that the suppression impacts bipolar cell release (*Figure 2A*). Note that the divisive gain control matches earlier models of both presynaptic inhibition (*Olsen and Wilson, 2008*) and synaptic depression (*Markram et al., 1998*). The suppressive term drops below one when the stimulus matches the suppressive filter, causing a proportional decrease in excitation of the ganglion cell. If the suppression does not contribute to the response, its nonlinearity would simply maintain a value of one, and the DivS model reduces to the LN model. The DivS model construction can be tractably fit to data using recent advances in statistical modeling (*Ahrens et al., 2008b*; *McFarland et al., 2013*).

The DivS model fits were highly consistent across the population, with similarly shaped excitatory and suppressive filters across cells (*Figure 2B*). For each cell, the suppressive filter was delayed relative to the excitatory filter ($10.9 \pm 2.2$ ms, $p < 0.0005$, $n = 13$, *Figure 2C*). The excitatory nonlinearity was approximately linear over the range of stimuli (*Figure 2D*, *left*), whereas the suppressive nonlinearity decreased below one when the stimulus either matched or was opposite to the suppressive filter (*Figure 2D*, *right*), resulting in 'ON-OFF' selectivity to both light increments and decrements.

The DivS model outperformed the LN model in predicting the observed currents (*Figure 2E*). Furthermore, it performed as well or better than models with other nonlinear interactions between the two filters. We first tested a more general form of nonlinear interaction by directly estimating a two-dimensional nonlinear function based on the filters derived from the DivS model. This 2-D nonlinearity maps each combination of the excitatory and suppressive filter outputs to a predicted current (*Figure 2F*; see Materials and methods). While this 2-D model contains many more parameters than the DivS model, it did not perform significantly better (*Figure 2E*); indeed, the estimated 2-D nonlinearities for each neuron were well approximated by the separable mathematical form of the DivS model ($R^2$ for 2-D nonlinearity reconstruction $= 0.94 \pm 0.02$; *Figure 2G*). We also tested an additive

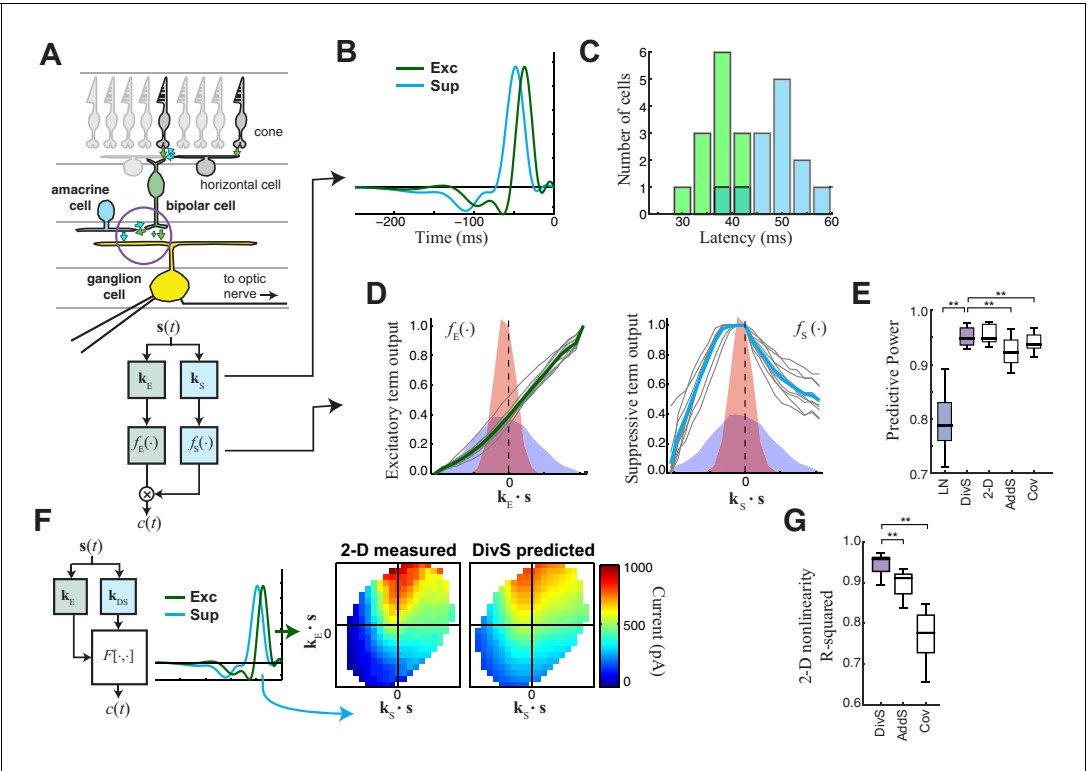

**Figure 2.** The divisive suppression (DivS) model of synaptic currents. (**A**) Schematic of retinal circuitry. The vertical excitatory pathway, cones → bipolar cells → ganglion cell, can be modulated at the bipolar cell synapse by amacrine cell-mediated inhibition of bipolar cell release or by synaptic depression. We model both processes by divisive suppression (*bottom*), where an LN model, representing the collective influence of amacrine cell inhibition and synaptic depression, multiplicatively modulates excitatory inputs from bipolar cells to the ganglion cell. (**B**) The excitatory (green) and suppressive (cyan) temporal filters of the DivS model for an example ON-Alpha cell. (**C**) Divisive suppression is delayed relative to excitation, demonstrated by the distributions of latencies measured for each pair of filters (mean delay = 10.9 ± 2.2 ms, p<0.0005, n = 13). (**D**) Excitatory (*left*) and suppressive nonlinearities (*right*) for the DivS model. The solid line indicates model fits for the example cell, and the gray lines are from other cells in the population, demonstrating their consistent form. The distribution of the filtered stimulus is also shown as the shaded area for HC (blue) and LC (red). The suppressive nonlinearity (*right*) falls below one for stimuli that match the kernel or are opposite, implying that divisive suppression is ON-OFF. (**E**) To validate the form of the DivS model, we compared its performance to alternative models, including a more general model where the form of the nonlinearity is not assumed (2-D, see below), a model where excitatory and suppressive terms interact additively (AddS) instead of divisively, and a covariance (COV) model similar to spike triggered covariance (*Figure 2—figure supplement 1*). The DivS model performed significantly better than the LN, AddS and COV models (**p<0.0005, n = 13), and matched the performance of the 2-D model. (**F**) We used a 2-dimensional nonlinearity to capture any more general interaction between excitatory and suppressive filters, shown with schematic (*left*), and the resulting fits (*middle*). Consistent with the model performance (**E**), the form of this 2-D nonlinearity could be reproduced by the DivS model (*right*). (**G**) Accuracy of the ability of the DivS, AddS, and COV models to reproduce the 2-D nonlinearity across neurons (**p<0.0005, n = 13).

The following figure supplement is available for figure 2:

**Figure supplement 1.** Comparison to covariance-based models.

suppression (AddS) model, where suppression interacts with excitation additively (see Materials and methods). The AddS model had significantly worse predictive power than the DivS model (p<0.0005, n = 13; *Figure 2E*) and less resemblance to the corresponding 2-D nonlinearities compared to the DivS model (p<0.0005, n = 13; *Figure 2G*).

Finally, we compared the DivS model to a form of spike-triggered covariance (*Fairhall et al., 2006*; *Liu and Gollisch, 2015*; *Samengo and Gollisch, 2013*) adapted to the continuous nature of the synaptic currents (see Materials and methods). This covariance analysis generated different filters than the DivS model (*Figure 2—figure supplement 1*), although both sets of filters were within the same subspace (*Butts et al., 2011*; *McFarland et al., 2013*), meaning that the covariance-based filters could be derived as a linear combination of the DivS filters and vice versa. Because the filters

shared the same subspace, the 2-D nonlinear mapping that converts the filter output to a predicted current had roughly the same performance as the 2-D model based on the DivS filters (*Figure 2E*). However, because the covariance model used a different pair of filters (and in particular the DivS filters are not orthogonal), its 2-D mapping differed substantially from that of the DivS model. Consequently, the 2-D mapping for the STC analysis, unlike the DivS analysis, could not be decomposed into two 1-D components (*Figure 2—figure supplement 1*) (*Figure 2G*). Thus, despite the ability of covariance analysis to nearly match the DivS model in terms of model performance (*Figure 2E*), it could not reveal the divisive interaction between excitation and suppression.

The DivS model therefore provides a parsimonious description of the nonlinear computation at the bipolar-ganglion cell synapse and yields interpretable model components, suggesting an interaction between tuned excitatory and suppressive elements. As we demonstrate below, the correspondingly straightforward divisive interaction detected by the DivS model on the ganglion cell synaptic input is essential in deriving the most accurate model of ganglion cell output, which combines this divisive interaction with subsequent nonlinear components related to spike generation.

## Divisive suppression explains contrast adaptation in synaptic currents

In addition to nearly perfect predictions of excitatory current at high contrast (*Figure 2*; *Figure 3C*), the DivS model also predicted the time course of the synaptic currents at low contrast. Indeed, using a single set of parameters, the model was similarly accurate in both contrast conditions (*Figure 3A*), and outperformed an LN model that used separate filters fit to each contrast level (e.g., *Figure 1E*). The DivS model thus implicitly adapts to contrast with no associated changes in parameters.

The adaptation of the DivS model arises from the scaling of the divisive term with contrast. The fine temporal features in the synaptic currents observed at high contrast (*Figure 3C*, *left*) arise from the product of the output of the excitatory LN component and the output of the suppressive LN component. Because suppression is delayed relative to the excitation and has both ON and OFF selectivity, suppression increases at both positive and negative peaks of the suppressive filter output (*Figure 3C inset*). This divisive suppression makes the DivS model output more transient compared to its excitatory component output alone; the difference between the two predictions is pronounced surrounding the times of peak excitation. At low contrast (*Figure 3C*, *right*), both excitatory and suppressive filter outputs are proportionately scaled down. Because the suppression is divisive and close to one, the DivS model becomes dominated by the excitatory term and closely matches the LN model, as well as the measured excitatory current.

The close match between data and DivS predictions across contrast levels suggests that the DivS model should exhibit contrast adaptation, as measured by the LN model filters (*e.g.*, *Figure 1E*). Indeed, using LN analysis to describe the DivS-model-predicted currents across contrast shows that the changes in filtering properties predicted by the DivS model were tightly correlated with those from the measured data (*Figure 3D*), including changes in contrast gain and biphasic index (*Figure 3E*). Furthermore, a small tonic offset of the synaptic currents across contrast levels (*Figure 1—figure supplement 1*), which resulted in a vertical shift in the nonlinearity of the LN model (*Figure 1E*; *Figure 1—figure supplement 1*), was captured by the DivS model without any parameter changes (*Figure 3F*).

## Divisive suppression largely originates from the surround region of the receptive field

The mathematical form of the DivS model (*Figure 2A*) is consistent with two pre-synaptic mechanisms that shape temporal processing: synaptic depression (*Jarsky et al., 2011*; *Ozuysal and Baccus, 2012*) and presynaptic inhibition (*Eggers and Lukasiewicz, 2011*; *Schubert et al., 2008*). Indeed, a model of ganglion cells that explicitly implements synaptic depression, the linear-nonlinear-kinetic model (LNK model) (*Ozuysal and Baccus, 2012*) can also predict the temporal features of ganglion cell intracellular recordings across contrast. The LNK model fits a single LN filter (analogous to the excitatory $\mathbf{k}_E$ and $f_E(.)$ of the DivS model; *Figure 2A*), with additional terms that simulate use-dependent depletion of output (*Figure 4—figure supplement 1*). This depletion depends on the previous output of the model (recovering over one or more time scales), and divisively modulates the output of the LN filter. For our data, the LNK model captured excitatory currents in response to the temporally modulated spot (*Figure 4A*), also outperforming the LN model (p<0.0005, *n* = 13),

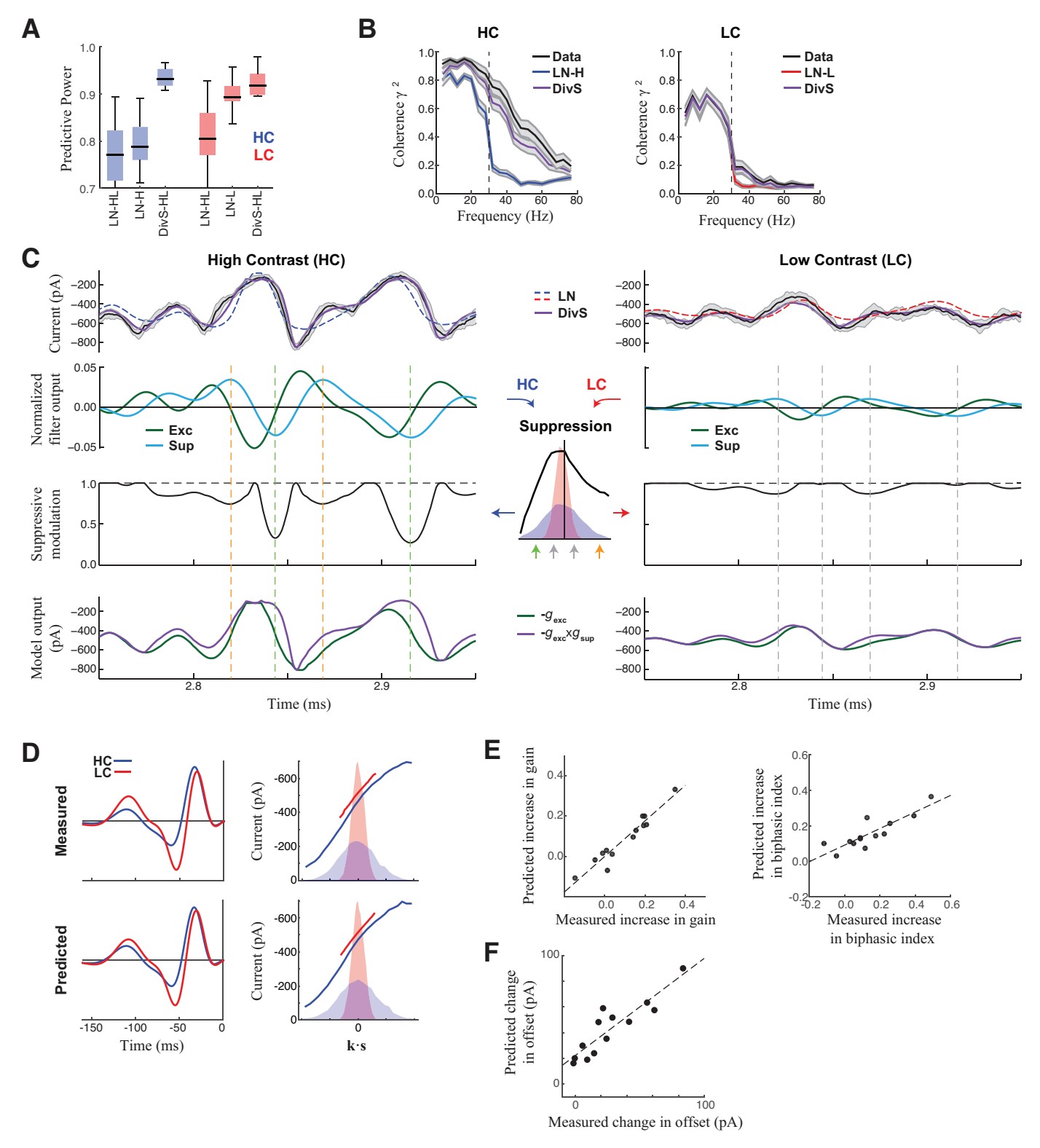

**Figure 3.** DivS model explains temporal precision and contrast adaptation in synaptic currents. (**A**) The predictive power of models across contrasts. The DivS model is fit to both contrasts using a single set of parameters, and outperforms LN models fit separately to either high or low contrast (LN-H and LN-L). As expected, the LN model fit for both contrasts (LN-HL) performs worse than separately fit LN models, because the LN-HL model cannot capture the filter changes without changes in model parameters. (**B**) Average coherence between model predictions and recorded synaptic currents on

*Figure 3 continued*

individual trials (n = 13), shown for high contrast (HC) and low contrast (LC). The DivS model prediction performs almost identically to the trial-averaged response. (C) DivS model explains precision and contrast adaptation through the interplay of excitation and suppression. *Top*: comparison of predictions of synaptic current response of the LN model and the DivS model for the cell in *Figure 1. second row*: normalized output of the excitatory and delayed suppressive filter. *3rd row*: suppressive modulation obtained by passing the filtered output through the suppressive nonlinearity (*middle inset*). *Bottom*: excitatory output of the DivS model before and after the suppressive modulation. In LC, the suppressive term (*third row*) does not deviate much from unity, and consequently the DivS model output resembles the excitatory input. (D) Comparison of the measured (*left*) and DivS model predicted (*right*) LN models across contrast. (E) The LN analysis applied to the DivS model predictions captures changes of both contrast gain (*left*: R = 0.96, p<10⁻⁶) and biphasic index (*right*: R = 0.86, p<0.0005) of the temporal filters across contrasts. (F) The DivS models predict the changes in tonic offset without any additional parameter shifts (R = 0.90, p<10⁻⁴).

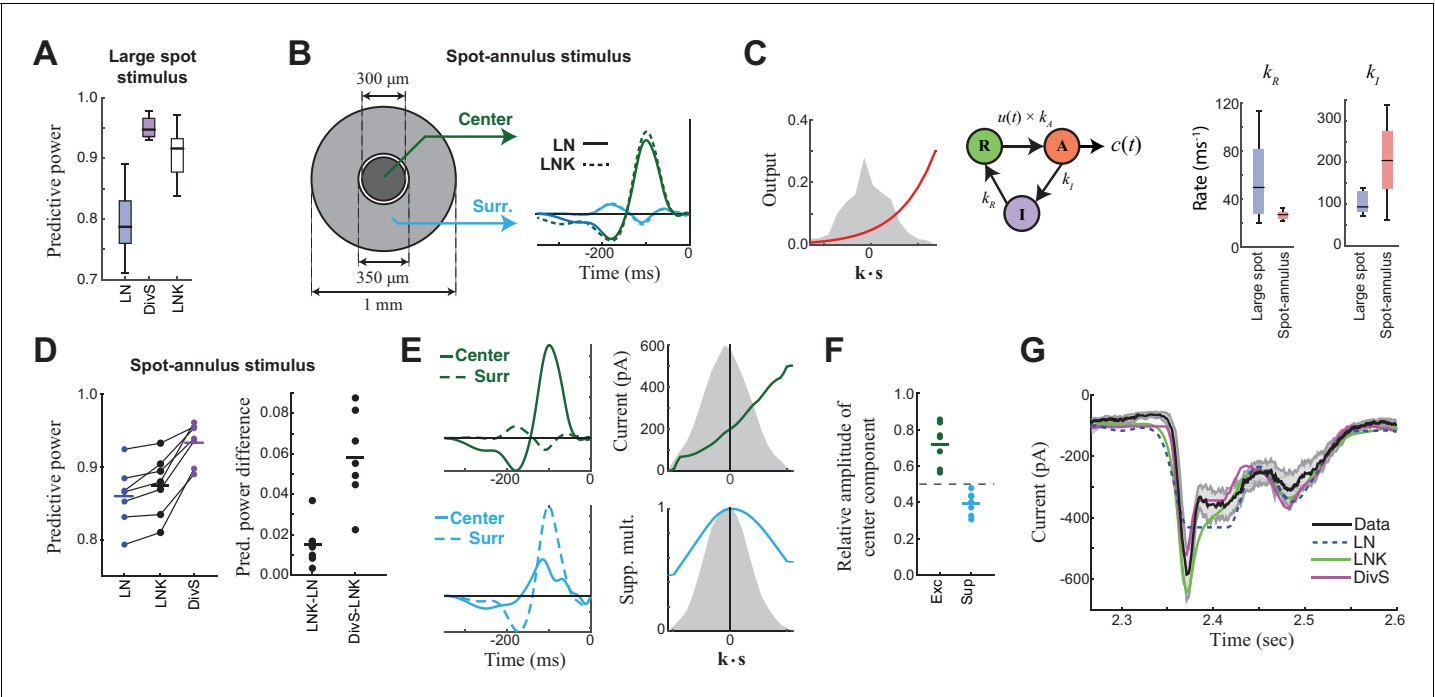

**Figure 4.** Probing the mechanism of divisive suppression with center-surround stimuli. (A) For the large spot stimulus, the Linear-Nonlinear-Kinetic (LNK) model nearly matches the performance of the DivS model, and outperforms the LN model. (B) To distinguish between different sources of divisive suppression, we presented a spot-annulus stimulus (*left*), where each region is independently modulated. Model filters can be extended to this stimulus using a separate temporal kernel for center and surround, shown for the LN and LNK model filters (*right*), which are very similar. (C) After the linear filter, the LNK model applies a nonlinearity (*left*), whose output drives the transition between resting and activated states (*middle*), which is further governed by kinetics parameters as shown. Critical kinetics parameters for LNK models differed between the large-spot and spot-annulus stimulus (*right*), with the spot-annulus model very quickly transitioning from Inactive back to Active states, minimizing the effects of synaptic depression. (D) The performance of the spatiotemporal LNK model is only slightly better than that of the LN model, and neither captures the details of the modulation in synaptic current, compared with the DivS model. (E) The spatiotemporal DivS model shown for an example neuron exhibits different spatial footprints for excitation and suppression, with excitation largely driven by the spot and suppression by the annulus. This divisive suppression cannot be explained exclusively by synaptic depression, which predicts overlapping sources of suppression and excitation (*Figure 4—figure supplement 1* and *2*). (F) The contribution of the center component in the DivS model for excitation (*left*) and suppression (*right*). Excitation was stronger in the center than in the surround (center contribution>0.5, p=0.016, n = 7) and suppression was weaker in the center (center contribution<0.5, p=0.016, n = 7) for every neuron. (G) The DivS model captured temporal transients in the current response to spot-annulus stimuli better than the LN and LNK models.

The following figure supplements are available for figure 4:

**Figure supplement 1.** DivS model localizes the suppressive components of LNK model and reproduces its simulated response.

**Figure supplement 2.** DivS model descriptions of extended LNK models.

although not with the level of performance as the DivS model (p<0.0005, *n* = 13). Furthermore, when data were generated de novo by an LNK model simulation, the resulting DivS model fit showed a delayed suppressive term, whose output well approximated the effect of synaptic depression in the LNK model (*Figure 4—figure supplement 1*).

The DivS and LNK models, however, yielded distinct predictions to a more complex stimulus where a central spot and surrounding annulus were modulated independently (*Figure 4B*). The models described above were extended to this stimulus by including two temporal filters, one for the center and one for the surround. As expected from the center-surround structure of ganglion cell receptive fields, an LN model fit to this condition demonstrated strong ON-excitation from the center, and a weaker OFF component from the surround (*Figure 4B*).

The 'spatial' LNK model's filter resembled that of the LN model (*Figure 4B*). Consistent with this resemblance to the LN Model, the spatial LNK model had rate constants that minimized the time that the model dwelled in the inactivated state (i.e., was 'suppressed') (*Figure 4C*). These rate constants were significantly different from those of the LNK model fit to the single temporally modulated spot. Correspondingly, the LNK model in the spot-annulus condition exhibited little performance improvement over the LN model (predictive power improvement 1.8% ± 1.3%, p=0.016, *n* = 7; *Figure 4D*).

By comparison, the DivS model significantly outperformed the LN model with an improvement of 8.6% ± 3.3% (p=0.016; *n* = 7), and was 6.7 ± 2.8% better than the LNK model (p=0.016; *n* = 7). The suppressive term of the DivS model showed a very distinct spatial profile relative to excitation, with a greater drive from the annulus region, while excitation was mostly driven by the spot region (*Figure 4E,F*). The suppressive filter processing the spot region was typically slower than the annulus filter: the peak latency for the suppressive filter was 129 ± 16 ms within the spot region compared to 120 ± 15 ms within the annulus region (faster by 9.7 ± 4.3 ms; p=0.0156, *n* = 7).

The strong suppression in the surround detected by the DivS model could not be explained by the LNK model, which cannot flexibly fit an explicit suppressive filter. Indeed, suppression in the LNK model arises from excitation, and thus the two components share the same spatial profile (*Figure 4B*; *Figure 4—figure supplement 1* and *2*). This can be demonstrated not only with simulations of the LNK model, but also more complex models with separate synaptic depression terms in center and surround (*Figure 4—figure supplement 2*). In all cases, application of the DivS model to data generated by these synaptic-depression-based simulations revealed that the suppressive term roughly matched the spatial profile of excitation, which is inconsistent with the observed data (*Figure 4F*). While these analyses do not eliminate the possibility that synaptic depression plays a role in shaping the ganglion cell response (and contributing to the suppression detected by the DivS model), the strength of surround suppression detected by the DivS model suggests that synaptic depression alone cannot fully describe our results.

## Nonlinear mechanisms underlying the spike output of ganglion cells

With an accurate model for excitatory synaptic currents established, we returned to modeling the spike output of ON-Alpha cells. Following previous likelihood-based models of ganglion cell spikes, we added a spike-history term, which implements absolute and relative refractory periods (*Butts et al., 2011*; *McFarland et al., 2013*; *Paninski, 2004*; *Pillow et al., 2005*). The output of this spike-history term is added to the output of the DivS model for the synaptic currents, and this sum is further processed by a spiking nonlinearity (*Figure 5A*) to yield the final predicted firing rate. Using a standard likelihood-based framework, all terms of the model – including the excitatory and suppressive LN models that comprised the prediction of synaptic currents – can then be tractably fit using spike data alone. But it is important to note that this model architecture was only made clear via the analyses of synaptic currents described above.

When fit using spiking data alone, the resulting excitatory and suppressive filters and nonlinearities closely resembled those found when fitting the model to the synaptic currents recorded from the same neurons (e.g., *Figure 2B,D*). Suppression was consistently delayed relative to excitation (*Figure 5B*), and exhibited both ON and OFF selectivity (*Figure 5C*). The spike-history term was suppressive and had two distinct components, a strong absolute refractory period that lasted 1–2 ms and a second relative refractory period that lasted more than 15 ms (*Berry and Meister, 1998*; *Butts et al., 2011*; *Keat et al., 2001*; *Paninski, 2004*; *Pillow et al., 2005*).

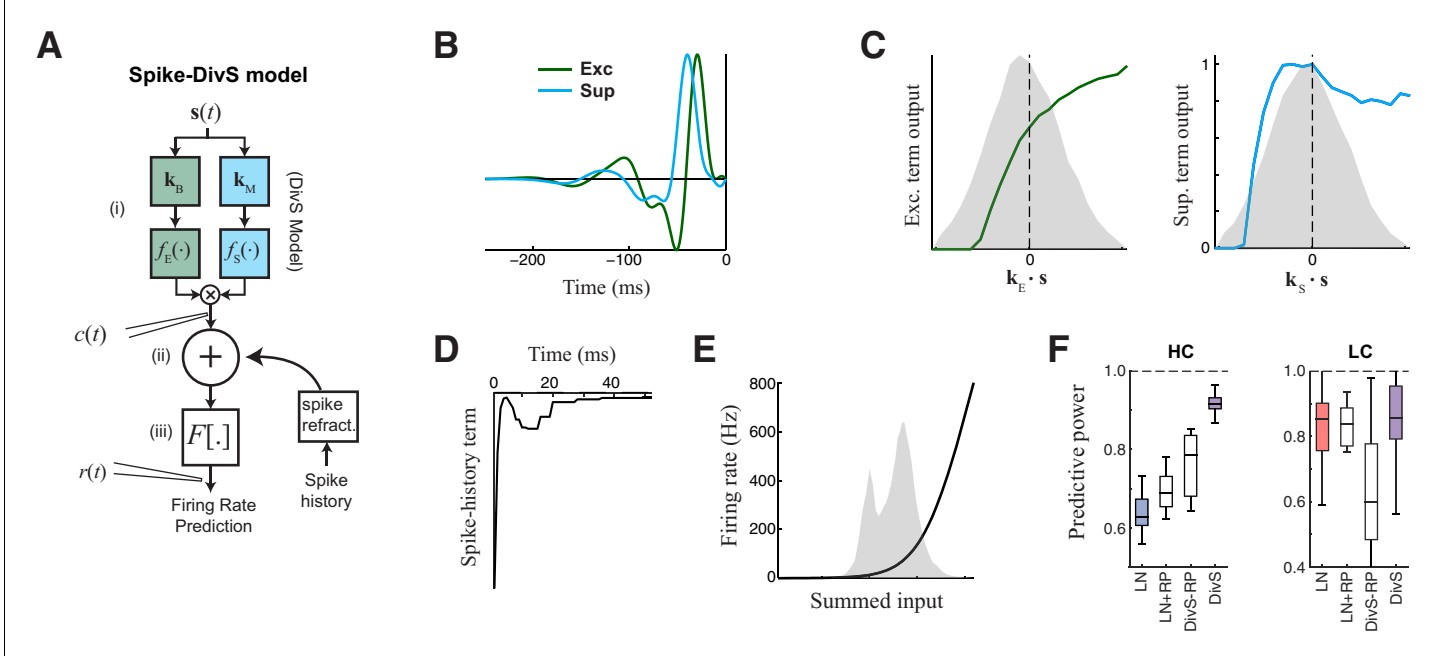

**Figure 5.** The extended divisive suppression model explains ganglion cell spike trains with high precision. (A) Model schematic for the divisive suppression model of spiking, which extends DivS model for the current data by adding an additional suppressive term for spike-history (refractoriness), with the resulting sum passed through a rectifying spiking nonlinearity. (B–E) The model components for the same example neuron considered in *Figures 1–3*. (B) The excitatory and suppressive filters. (C). The excitatory and suppressive nonlinearities. The filters and nonlinearities were similar to the DivS model fit from current data (shown in *Figure 2B*). (D) The spike-history term, demonstrating an absolute and relative refractory period. (E) The spiking nonlinearity, relative to the distribution of generating signals (shaded). (F). The predictive power of different models applied to the spike data in HC and LC. The DivS model performs better than other models (HC: p<0.001; LC: p<0.002, n = 11), including the LN model, the LN model with spike history term (LN+RP), and a divisive suppression model lacking spike refractoriness (DivS-RP). Only a single set of parameters was used to fit the DivS model for both contrasts, whereas all other models shown used different parameters fit to each contrast.

The resulting model successfully captured over 90% of the predictable variance in the firing rate for all neurons in the study (*Figure 5F*, median = 91.5% ± 1.0%; n = 11), representing the best model performance reported in the literature for ganglion cell spike trains considered at millisecond resolution. By comparison, the standard LN model had a median predictive power of 62.8% ± 1.9% (n = 11); which modestly increased to 68.8% ± 1.9% upon inclusion of a spike-history term (*Figure 5F*). This suggests that ganglion cell spikes are strongly shaped by the nonlinear computations present at their synaptic input, and that the precise timing of ganglion cell spiking involves the interplay of divisive suppression with spike-generating mechanisms.

## Precision of spike trains arises from complementary mechanisms of divisive suppression and spike refractoriness

To evaluate the relative contributions of divisive suppression and spike refractoriness to predicting firing, we simulated spike trains using different combinations of model components (*Figure 6A*). We found that the parameters of the divisive suppression components could not fit without including a spike-history term, suggesting that each component predicts complementary forms of suppression. We could generate a DivS model without a spike-history term, however, by first determining the full model (with spike-history term), and then removing the spike-history term and refitting (see Materials and methods), resulting in the DivS–RP model. This allowed for direct comparisons between models with a selective deletion of either divisive suppression or spike refractoriness (*Figure 6A*).

Event analyses on the resulting simulated spike trains, compared with the observed data, demonstrate that both divisive suppression (derived from the current analyses above) and spike refractoriness were necessary to explain the precision and reliability of ganglion cell spike trains. By

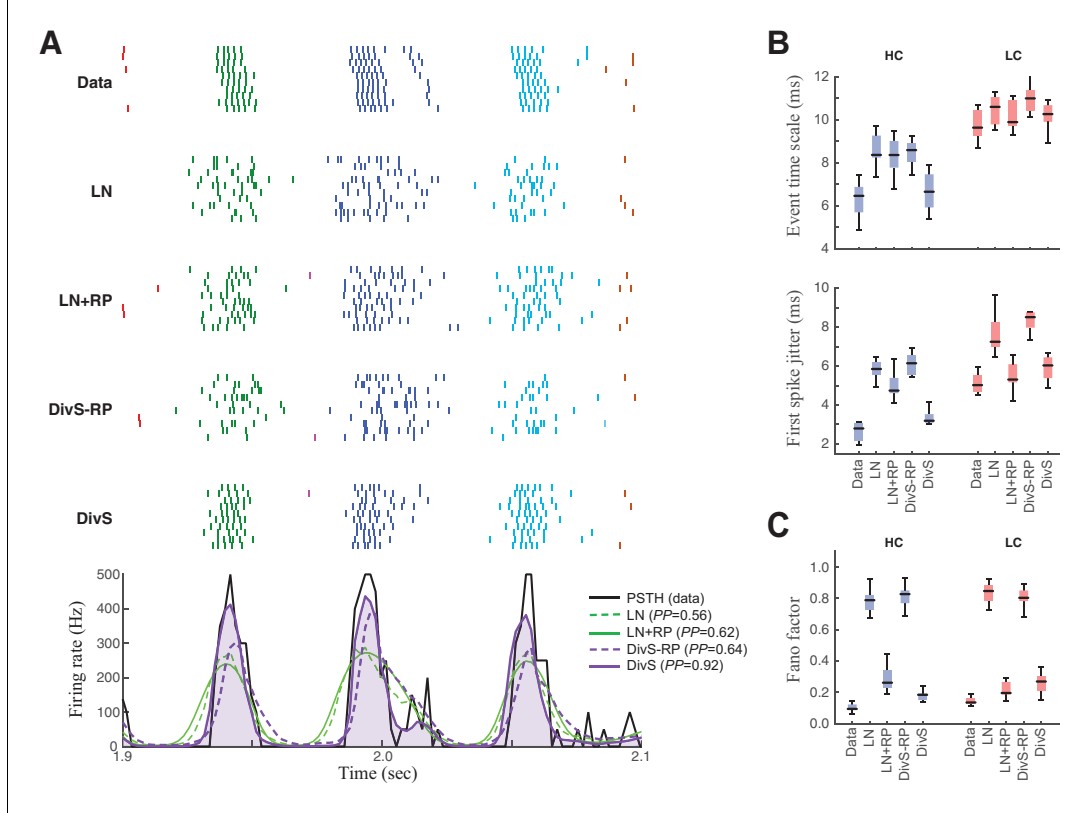

**Figure 6.** Spike patterning is shaped by a combination of nonlinear mechanisms. (**A**) *Top*: Spike rasters recorded over ten repeats for an example cell (black) compared with simulated spikes from four models: LN, LN model with spike-history term (LN+RP), the DivS model without spike-history (DivS-RP), and the full DivS model (DivS). Colors in the raster label separate spike events across trials (see Materials and methods). *Bottom*: The PSTHs for each model demonstrate that suppressive terms are important in shaping the envelope of firing (DivS prediction is shaded). (**B–E**) Using event labels, spike statistics across repeats were compiled to gauge the impact of different model components. (**B**) The temporal properties of events compared with model predictions, across contrast (same as *Figure 1*, with DivS-based models added). Both spike-history and divisive suppression contribute to reproduce the temporal scales across contrast. (**C**) The Fano factor for each event is a measure of reliability, which increased (i.e., Fano factor decreased) for models with a spike-history term.

comparing the two models without DivS (LN and LN+RP) to those with DivS (DivS and DivS–RP), it is clear that divisive suppression is necessary to predict the correct envelope of the firing rate (*Figure 6B*). Note, however, that DivS had little impact in the low contrast condition, which lacked fine-time-scale features of the spike response.

By comparison, the spike-history term had little effect on the envelope of firing (*Figure 6A*, *bottom*), and contributed little to the fine time scales in the ganglion cell spike train at high contrast (*Figure 6B*). Instead, the spike-history term had the largest effect on accurate predictions of event reliability, as reflected in the event Fano factor (*Figure 6C*). By comparison, both models without the spike-history term had much greater variability in spike counts within each event. The presence of the suppression contributed by the spike-history term following each event allows the predicted firing rate to be much higher (and more reliable) during a given event, resulting in reliable patterns of firing within each event (*Figure 6A*) (*Pillow et al., 2005*).

We conclude that a two-stage computation present in the spike-DivS model, with both divisive suppression and spike refractoriness, is necessary to explain the detailed spike patterning on ON-Alpha ganglion cells.

## Enhancement of contrast adaptation via spike refractoriness in ganglion cell output

In addition to accurate reproduction of precise spike outputs of ganglion cells, the DivS model also captured the effects of contrast adaptation observed in the ganglion cell spike trains. For both contrast conditions, the simulated spike trains, which are predicted for both contrasts using a single set of parameters, were almost indistinguishable from the data (*Figure 7A*, *top*). As with the performance of the models of excitatory current (*Figure 3*), the DivS model outperformed LN models that were separately fit for each contrast level (*Figure 5F*).

The ability to correctly predict the effects of contrast adaptation depended on both the divisive suppression and spike-refractoriness of the spike-DivS model. This is shown for an example neuron by using LN filters of the simulated output of each model at high and low contrasts (*Figure 7B*). In this case, only the DivS model (which includes a spike-history term) shows adaptation similar to that observed by the LN filters fit to the data. We quantified this by identifying the most prominent feature of adaptation of the LN filters, the change in filter amplitude (i.e., contrast gain). Across the population, the DivS correctly predicted the magnitude of this change (*Figure 7C*, *top*), as well as the changes in biphasic index across contrasts (*Figure 7C*, *bottom*), and outperformed models with either the divisive suppression or spike-history terms missing.

As expected, spike refractoriness imparted by the spike-history term contributed to the stronger effects of contrast adaptation observed in spikes relative to synaptic inputs (*Beaudoin et al., 2007*; *Kim and Rieke, 2001*, *2003*; *Rieke, 2001*; *Zaghloul et al., 2005*). Specifically, at high contrast, spikes concentrate into relatively smaller time windows, leading to a consistently timed effect of spike refractoriness (*Figure 7D*). As a result, despite similar numbers of spikes at the two contrasts, the effect of the spike-history term has a bigger impact at high contrast.

Thus, fast contrast adaptation – and more generally the temporal shaping of ON-Alpha ganglion cell spike trains – depends on nonlinear mechanisms at two stages of processing within the retinal

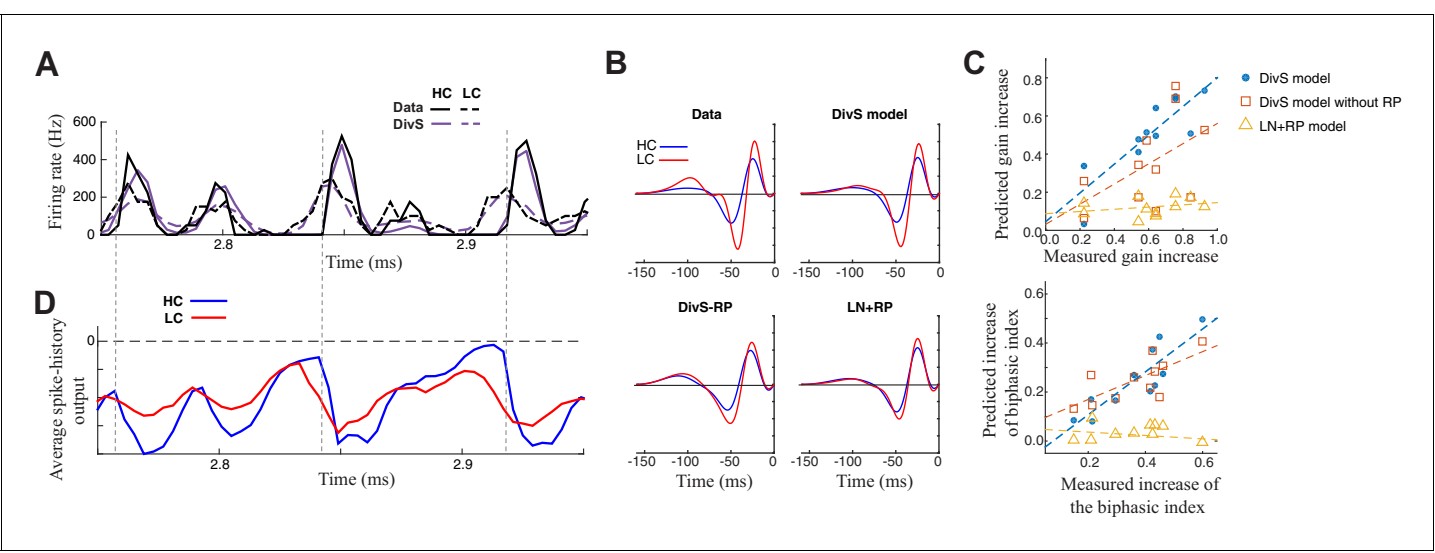

**Figure 7.** Contrast adaptation in the spike output depends on both divisive suppression and spike refractoriness. (A) The full spike-DivS model accurately captured contrast adaptation. *Top*: observed PSTH and predicted firing rates of the DivS model at HC and LC. (B) The DivS model predicted the changes in LN filter shape and magnitude with contrast for an example cell. Predicted changes are shown for each model, demonstrating that the full effects of contrast adaptation require both divisive suppression and spike-history terms. (C) Measured and predicted contrast gain (*top*) and changes of biphasic index (*bottom*). The DivS model accurately predicted a contrast gain and changes biphasic index across contrast across cells (contrast gain: slope of regression = 0.75, *R* = 0.85, p<0.001; biphasic index: slope of regression = 0.87, *R* = 0.87, p<0.001). DivS model without the spike history term underestimated contrast adaptation (contrast gain: slope of regression = 0.53, *R* = 0.51, p = 0.10; biphasic index: slope of regression = 0.49, *R* = 0.73, p<0.05), and the LN+RP model failed to predict adaptation altogether (contrast gain: slope of regression = 0.06, *R* = 0.28, p = 0.41; biphasic index: slope of regression = −0.07, *R* = −0.18, p = 0.60). (D) The suppressive effect from the spike-history term was amplified at HC, due to the increased precision of the spike train. Dashed lines show the onset of HC spike events, which predict the largest difference in the magnitudes of the suppression between contrasts.

circuit. Both aspects of nonlinear processing originate at the level of ganglion cell synaptic inputs, shaped by divisive suppression, and become amplified by spike-refractoriness to generate the array of nonlinear properties evident in the spike train output.

## Discussion

In this study, we derived a retina-circuit-inspired model for ganglion cell computation using recordings of both the synaptic inputs and spike outputs of the ON-Alpha ganglion cell. Data were used to fit model parameters and evaluate different hypotheses of how the retinal circuit processed visual stimuli. The resulting model explained both high precision firing and contrast adaptation with unprecedented accuracy. Precise timing was already present in the excitatory synaptic inputs, and can be explained by divisive suppression, which likely depends on a combination of mechanisms: presynaptic inhibition of bipolar terminals from amacrine cells and synaptic depression at bipolar cell synapses. The interplay between nonlinear mechanisms, including divisive suppression, spike refractoriness and spiking nonlinearity, accurately captured the detailed structure in the spike response across contrast levels.

Divisive suppression was implemented by multiplying two LN models together (*Figure 2A*). One LN model controls the gain of a second LN model; the gain is equal to or less than one and so represents division. While divisive gain terms have been previously suggested in the retina – particularly in reference to early models of contrast adaptation (*Mante et al., 2008*; *Meister and Berry, 1999*; *Shapley and Victor, 1978*), critical novel elements of the present DivS model include the ability to fit the nonlinearities of both LN terms by themselves, as well as their tractability in describing data at high time resolution. The presence of nonlinearities that are fit to data in the context of multiplicative interactions distinguishes this model from multi-linear models (i.e., two linear terms multiplying) (*Ahrens et al., 2008a*; *Williamson et al., 2016*), as well as more generalized LN models such as those associated with spike-triggered covariance (*Fairhall et al., 2006*; *Samengo and Gollisch, 2013*; *Schwartz et al., 2006*). Furthermore the model form allows for inclusion of spike-history terms as well as spiking nonlinearities, and can be tractably fit to both synaptic currents and spikes at high temporal resolution (~1 ms).

An eventual goal of our approach is to characterize the nonlinear computation performed on arbitrarily complex spatiotemporal stimuli. Here, we focused on temporal stimuli, which drive well-characterized nonlinearities in ganglion cell processing including temporal precision (*Berry and Meister, 1998*; *Butts et al., 2007*; *Keat et al., 2001*; *Passaglia and Troy, 2004*; *Uzzell and Chichilnisky, 2004*) and contrast adaptation (*Kim and Rieke, 2001*; *Meister and Berry, 1999*; *Shapley and Victor, 1978*) but do not require a large number of additional parameters to specify spatial tuning. By comparison, studies that focused on characterizing nonlinearities in spatial processing (*Freeman et al., 2015*; *Gollisch, 2013*; *Schwartz and Rieke, 2011*) have not modeled responses at high temporal resolution. Ultimately, it will be important to combine these two approaches, to capture nonlinear processing within spatial 'subunits' of the ganglion cell receptive field, and thereby predict responses at both high temporal and spatial resolutions to arbitrary stimuli. Such an approach would require a large number of model parameters and consequently a larger amount of data than collected here. Our intracellular experiments were useful for deriving model architecture – discerning the different time courses of excitation, suppression, and spike refractoriness – but ultimate tests of full spatiotemporal models will likely require prolonged, stable recordings of spike trains, perhaps using a multielectrode array.

### Generation of temporal precision in the retina

One important nonlinear response property of early sensory neurons is high temporal precision. Temporal precision of spike responses has been observed in the retinal pathway with both noise stimuli (*Berry et al., 1997*; *Reinagel and Reid, 2000*) and natural movies (*Butts et al., 2007*). The precise spike timing suggests a role for temporal coding in the nervous system (*Berry et al., 1997*), or alternatively simply suggests that analog processing in the retina must be oversampled in order to preserve information about the stimulus (*Butts et al., 2007*). Temporal precision also plays an important role in downstream processing of information provided by ganglion cells (*Stanley et al., 2012*; *Usrey et al., 2000*).

The generation of temporal precision involves nonlinear mechanisms within the retina, which may include both spike-refractoriness within ganglion cells (*Berry and Meister, 1998*; *Keat et al., 2001*; *Pillow et al., 2005*) and the interplay of excitation and inhibition (*Baccus, 2007*; *Butts et al., 2016*; *Butts et al., 2011*). Such distinct mechanisms contributing to ganglion cell computation are difficult to distinguish using recordings of the spike outputs alone, which naturally reflect the total effects of all upstream mechanisms. By recording at two stages of the ganglion cell processing, we demonstrated that high temporal precision has already presented in the synaptic current inputs at high contrast, and temporal precision of both current inputs and spike outputs can be accurately explained by the DivS model.

The DivS model explained fast changes in the response through the interplay of excitation and suppression. For both the spike and current models, suppression is consistently delayed relative to excitation. The same suppression mechanism also likely underlies high temporal precision of LGN responses, which can be captured by a model with delayed suppression (*Butts et al., 2011*). Indeed, recordings of both LGN neurons and their major ganglion cell inputs suggest that precision of LGN responses is inherited from the retina and enhanced across the retinogeniculate synapse (*Butts et al., 2016*; *Carandini et al., 2007*; *Casti et al., 2008*; *Rathbun et al., 2010*; *Wang et al., 2010b*). Therefore, our results demonstrate that the temporal precision in the early visual system likely originates from nonlinear processing in the inputs to retinal ganglion cells. Note that the full spiking-DivS model did not incorporate any form of direct synaptic inhibition onto the ON-Alpha cell, consistent with findings that the impact of such inhibition is relatively weak and that excitation dominates the spike response in the stimulus regime that we studied (*Kuo et al., 2016*; *Murphy and Rieke, 2006*).

Our results show that the contribution of spike-history term to precision – as measured by the time scale of events and first-spike jitter – seems minor, consistent with earlier studies in the LGN (*Butts et al., 2016*; *Butts et al., 2011*). Nevertheless, the spike-history term does play an important role in spike patterning within the event (*Pillow et al., 2005*) and the resulting neuronal reliability (*Berry and Meister, 1998*). In fact, we could not fit the divisive suppression term robustly without the spike-history term in place, suggesting that both nonlinear mechanisms are important to explain the ganglion cell firing.

## Contrast adaptation relies on both divisive suppression and spike refractoriness

Here we modeled contrast adaptation at the level of synaptic currents and spikes from the same ganglion cell. We found contrast adaptation in synaptic inputs to ganglion cells, consistent with previous studies (*Beaudoin et al., 2007*; *Kim and Rieke, 2001*; *Rieke, 2001*; *Zaghloul et al., 2005*). Such adaptation could be explained by divisive suppression, which takes a mathematical form similar to previously proposed gain control models (*Heeger, 1992*; *Shapley and Victor, 1979*). Because the suppressive nonlinearity has a very different shape than the excitatory nonlinearity, divisive suppression has a relatively strong effect at high contrast and results in a decrease in measured gain. Moreover, the same divisive suppression mechanism may also explain nonlinear spatial summation properties of ganglion cells (*Shapley and Victor, 1979*), because suppression generally has broader spatial profiles than excitation.

Contrast adaptation is amplified in the spike outputs mostly due to spike refractoriness and changes of temporal precision across contrast. At high contrast, the response had higher precision and occurred within shorter event windows (*Butts et al., 2010*). As a result, the accumulated effect of spike refractoriness was stronger within each response event. Note that the effect of the spike-history term was highly dependent on the ability of the model to predict fine temporal precision at high contrast, which largely originates from the divisive suppression term as discussed earlier. Therefore, the two nonlinear properties of retinal processing, contrast adaptation and temporal precision, are tightly related and can be simultaneously explained by the DivS model.

## Circuits and mechanisms underlying the divisive suppression

Divisive suppression has been observed in the invertebrate olfactory system (*Olsen and Wilson, 2008*), the lateral geniculate nucleus (*Bonin et al., 2005*), the primary visual cortex (*Heeger, 1992*), and higher visual areas such as area MT (*Simoncelli and Heeger, 1998*). A number of biophysical

and cellular mechanisms for divisive suppression have been proposed, including shunting inhibition (*Abbott et al., 1997*; *Carandini et al., 1997*; *Hao et al., 2009*), synaptic depression (*Abbott et al., 1997*), presynaptic inhibition (*Olsen and Wilson, 2008*; *Zhang et al., 2015*) and fluctuation in membrane potential due to ongoing activity (*Finn et al., 2007*).

We evaluated different mechanistic explanations of the divisive suppression identified in this study. Divisive suppression underlying synaptic inputs to ganglion cells cannot be attributed to fluctuations in membrane potential or shunting inhibition since we recorded synaptic currents under voltage-clamp conditions that minimize inhibitory inputs. Although synaptic depression could also explain fast transient responses and contrast adaptation (*Ozuysal and Baccus, 2012*), synaptic depression will generally result in excitation and suppression that have the same spatial profiles (*Figure 4—figure supplement 1* and *2*), whereas we show that excitation and suppression have distinct spatial profiles (*Figure 4*). Therefore, the divisive suppression in our model likely depends partly on presynaptic inhibition from amacrine cells, which can extend their suppressive influence laterally (*Euler et al., 2014*; *Franke et al., 2016*; *Schubert et al., 2008*). This was somewhat surprising, because earlier studies had shown that contrast adaptation persisted in the presence of inhibitory receptor antagonists, suggesting that adaptation depended primarily on mechanisms intrinsic to the bipolar cell (e.g., synaptic depression), independent of synaptic inhibition (*Beaudoin et al., 2007*; *Brown and Masland, 2001*; *Rieke, 2001*). Indeed, synaptic depression was likely the primary mechanism for adaptation in conditions with inhibition blocked, but the present results suggest that lateral inhibitory mechanisms also play a role in generating adaptation under conditions with inhibition intact, at least for some retinal circuits.

Models for contrast adaptation based on synaptic depression rely on a change in the average level of synaptic activity with contrast. This condition is met for synapses with lower rates of tonic release (*Jarsky et al., 2011*). However, the ON-Alpha cell receives a relatively high rate of tonic release from presynaptic type 6 bipolar cells (*Borghuis et al., 2013*; *Schwartz et al., 2012*). Consequently, the average excitatory synaptic input would change less during contrast modulation compared to a ganglion cell that received a lower rate of glutamate release. An inhibitory mechanism for contrast adaptation thus may play a relatively prominent role for retinal circuits, such as the ON-Alpha cell, driven by high rates of glutamate release.

Detailed anatomical studies suggest that each ganglion cell type receives inputs from a unique combination of bipolar and amacrine cell types, contributing to a unique visual computation (*Baden et al., 2016*). By focusing on a single cell type, the ON-Alpha cell, we identified a particular computation consistent across cells. We expect that other ganglion cell types will perform different computations, and likewise have different roles in visual processing. This could include additional contrast-dependent mechanisms, including slow forms of adaptation (*Baccus and Meister, 2002*; *Manookin and Demb, 2006*), sensitization (*Kastner and Baccus, 2014*) and complex changes in filtering (*Liu and Gollisch, 2015*). Thus, further applications of the approach described here will uncover a rich diversity of computation constructed by retinal circuitry to format information for downstream visual processing.

## Materials and methods

### Neural recordings

Data were recorded from ON-Alpha ganglion cells from the in vitro mouse retina using the procedures described previously (*Borghuis et al., 2013*; *Wang et al., 2011*). Spikes were recorded in the loose-patch configuration using a patch pipette filled with Ames medium, and synaptic currents were recorded using a second pipette filled with intracellular solution (in mM): 110 Cs-methanesulfonate; 5 TEA-Cl, 10 HEPES, 10 BAPTA, 3 NaCl, 2 QX-314-Cl, 4 ATP-Mg, 0.4 GTP-Na$_2$, and 10 phosphocreatine-Tris$_2$ (pH 7.3, 280 mOsm). Lucifer yellow was also included in the pipette solution to label the cell using a previously described protocol (*Manookin et al., 2008*). The targeted cell was voltage clamped at $E_{Cl}$ (−67 mV) to record excitatory currents after correcting for the liquid junction potential (−9 mV). Cells in the ganglion cell layer with large somas (20–25 μm diameter) were targeted. Cells were confirmed to be ON-Alpha cells based on previously established criteria (*Borghuis et al., 2013*): (1) an ON response; (2) high rate of spontaneous firing; and a high rate of spontaneous excitatory synaptic input; (3) a low input resistance (~40–70 MΩ). In some cases, we

imaged the morphology of recorded cells and confirmed (4) a relatively wide dendritic tree (300–400 µm diameter) and (5) stratification on the vitreal side of the nearby ON cholinergic (starburst) amacrine cell processes.

We made recordings from 27 ON-Alpha cells total, each in one or more of the experimental conditions described. Of the 15 cells recorded in cell-attached configuration (spike recordings), 4 cells were excluded where low reliability across trials indicated an unstable recording, as indicated by much higher spike event Fano Factors (>0.2, see below).

All procedures were conducted in accordance with National Institutes of Health guidelines under protocols approved by the Yale University Animal Care and Use Committee.

## Visual stimulation

The temporally modulated spot stimulus was described previously (*Wang et al., 2011*). The retina was stimulated by UV LEDs (peak, 370 nm; NSHU-550B; Nichia America) to drive cone photoreceptors in the ventral retina. UV LEDs were diffused and windowed by an aperture in the microscope's fluorescence port, with intensity controlled by *pClamp 9* software via a custom non-inverting voltage-to-current converter using operational amplifiers (TCA0372; ON Semiconductor). The stimulus was projected through a 4X objective lens (NA, 0.13). The stimulus was a flickering spot (1-mm diameter), with intensity generated from low pass Gaussian noise with a 30 Hz cutoff frequency. We used a contrast-switching paradigm (*Baccus and Meister, 2002*; *Kim and Rieke, 2001*; *Zaghloul et al., 2005*), in which the temporal contrast alternately stepped up or down every 10 s. The contrast of the stimulus is defined by the SD of the Gaussian noise and was either 0.3 times (high contrast) or 0.1 times (low contrast) the mean. Note that this is only a three-fold difference in contrast versus the seven-fold difference considered in *Ozuysal and Baccus (2012)*, but sufficient to see clear contrast effects. The stimulus comprised 10 cycles of 10 s for each contrast. The first 7 s were unique in each cycle (used for fitting model parameters), and the last 3 s were repeated across cycles (used for cross-validation of model performance).

The center-surround stimuli (*Figure 4B*) were generated in *Matlab* (Mathworks, Natick) using the *Psychophysics Toolbox* (*Brainard, 1997*) and presented with a video projector (M109s DLP; Dell, or identical HP Notebook Companion; HP), modified to project UV light (single LED NC4U134A, peak wavelength 385 nm; Nichia) as previously described (*Borghuis et al., 2013*). The center and surround stimuli were independently modulated with Gaussian noise (60-Hz update rate). A spot covered the receptive field center (e.g., 0.3 mm), and an annulus extended into the surround (e.g., inner/outer diameters of 0.35/1.0 mm). We recorded 7 ON-Alpha cells in this condition. For a subset of the recordings (*n*=5), we explored a range of inner/outer diameters, and selected the diameters that maximized the difference between the spatial footprints of excitatory and suppressive terms of the DivS model (see below).

The mean luminance of the stimulus was calculated to evoke ~$4 \times 10^4$ photoisomerizations cone$^{-1}$ sec$^{-1}$, under the assumption of a 1 µm$^2$ cone collecting area. For all methods of stimulation, the gamma curve was corrected to linearize output, and stimuli were centered on the cell body and focused on the photoreceptors. We verified that the relatively short stimulus presentation did not result in significant bleaching, as the response (and model parameters) had no consistent trends from the beginning of the experiment to the end (*Figure 1—figure supplement 1–2*).

## Statistical modeling of the synaptic current response

We modeled the synaptic current response of neurons using the traditional linear-nonlinear (LN) cascade model (*Paninski, 2004*; *Truccolo et al., 2005*), the Linear-Nonlinear-Kinetic model (*Ozuysal and Baccus, 2012*), a 2-D nonlinear model ('2-D'), and the Divisive Suppression model ('DivS') introduced in this paper.

In all cases (with the exception of the LN analyses of contrast adaptation effects described below), we optimized model parameters to minimize the mean-squared error (MSE) between the model-predicted and observed currents:

$$\mathrm{MSE} = \sum_t [c(t) - c_{\mathrm{obs}}(t)]^2 \tag{1}$$

To limit the number of model parameters in the minimization of MSE, we represented temporal filters by linear coefficients weighting a family of orthonormalized basis functions (*Keat et al., 2001*):

$$\zeta(t) = \sin[\pi n(2t/t_{\mathrm{F}} - (t/t_{\mathrm{F}})^2)], \tag{2}$$

where $t_{\mathrm{F}}$=200 ms.

## LN model

The LN model transforms the stimulus $\mathbf{s}(t)$ to the synaptic current response $c(t)$ using a linear filter $\mathbf{k}_{\mathrm{LN}}$ and nonlinearity $f_{\mathrm{LN}}[\cdot]$ such that:

$$c(t) = f_{\mathrm{LN}}[\mathbf{k}_{\mathrm{LN}} \cdot \mathbf{s}(t)] + c_0, \tag{3}$$

where $c_0$ is a baseline offset. The filter $\mathbf{k}_{\mathrm{LN}}$ was represented as a set of coefficients weighting the basis functions of *Equation 2*, and the nonlinearities were represented as coefficients weighting tent basis functions as previously described (*Ahrens et al., 2008b*; *McFarland et al., 2013*).

## 2-D model

We generalized the LN model by incorporating a second filtered input, such that:

$$c(t) = F[\mathbf{k}_{\mathrm{e}} \cdot \mathbf{s}(t), \mathbf{k}_{\mathrm{s}} \cdot \mathbf{s}(t)], \tag{4}$$

where $F[\cdot, \cdot]$ is a two-dimensional nonlinearity, and $\mathbf{k}_{\mathrm{e}}$ and $\mathbf{k}_{\mathrm{s}}$ denote the excitatory and suppressive filters respectively.

The 2-D nonlinearity was represented using piecewise planar surfaces and could be estimated non-parametrically for a given choice of filters (*Toriello and Velma, 2012*). Specifically, we divided the 2-D space into a set of uniform squares, and then subdivided each square into two triangles. Each basis function was defined as a hexagonal pyramid function centered at one of the vertices, and the 2-D nonlinearity function was expressed as a combination of these bases:

$$F[x, y] = \sum_{i,j} w_{ij} f_{ij}(x, y), \tag{5}$$

where $f_{ij}(x, y)$ is the basis centered at the $ij^{th}$ grid vertex, and $w_{ij}$ is the corresponding weight coefficient, which was optimized by minimizing MSE for a given choice of filters.

The coefficients for the filters and nonlinearities were optimized using block-coordinate descent: for a given choice of nonlinearity $F[\cdot, \cdot]$ the filters were optimized, and vice versa. In doing so, we introduced an additional constraint on the nonlinearity due to a degeneracy in the combined optimization of stimulus filters and 2-D nonlinearity. Specifically, one can choose a linear combination of the two stimulus filters and achieve the same model performance by refitting the 2-D nonlinearity. To alleviate this problem, we constrained the 2-D nonlinearity to be monotonically increasing along the first dimension, i.e.,

$$\text{If } x > x', \text{then } F[x, y] \geq F[x', y], \forall\, y \tag{6}$$

## DivS model

We derived the DivS model as a decomposition of the 2-D nonlinearity into two one-dimensional LN models that interact multiplicatively:

$$c(t) = f_{\mathrm{e}}[\mathbf{k}_{\mathrm{e}} \cdot \mathbf{s}(t)] \times f_{\mathrm{s}}[\mathbf{k}_{\mathrm{s}} \cdot \mathbf{s}(t)] + c_0 \tag{7}$$

We constrained the excitatory nonlinearity $f_{\mathrm{e}}(\cdot)$ to be monotonically increasing, and constrained the second nonlinearity $f_{\mathrm{s}}(\cdot)$ to be suppressive by bounding it between zero and one, with the value for zero input constrained to be one. We optimized the filters and the nonlinearities through block-coordinate descent until a minimum MSE was found. Because this optimization problem is in general non-convex (i.e., not guaranteed to have a single global minimum), we used standard approaches (*McFarland et al., 2013*) such as testing a range of initialization and block-coordinate descent procedures to ensure that global optima were found.

## Covariance model of currents and spikes

We performed covariance analyses on both synaptic current responses and spike trains. Spike-triggered covariance analyses followed established methods (*Fairhall et al., 2006*; *Liu and Gollisch, 2015*; *Samengo and Gollisch, 2013*; *Schwartz et al., 2006*). Briefly, for spike-triggered covariance analysis, we collected the stimulus sequence $s_n(\tau) = s(t_n - \tau)$ that preceded each spike time $t_n$, where the lag $\tau$ covers 200 time bins at 1 ms resolution. The spike-triggered average was calculated as the average over all spikes $\overline{s_{spk}}(\tau) = \langle s_n(\tau) \rangle_n$, and the covariance matrix was calculated as $C_{\tau_1,\tau_2}^{spk} = \langle [s_n(\tau_1) - \overline{s_{spk}}(\tau_1)][s_n(\tau_2) - \overline{s_{spk}}(\tau_2)] \rangle_n$. We then subtracted the 'prior' covariance matrix (averaged over all times instead of just spike times) from $C_{\tau_1,\tau_2}^{spk}$, and diagonalized the resulting 'covariance-difference' matrix to obtain its eigenvalues and corresponding eigenvectors.

This procedure was extended to perform the same analysis on synaptic currents, similar to past applications (*Fournier et al., 2014*). For current-based covariance analysis, we calculated the cross-correlation between stimulus and synaptic current response (analogous to the spike-triggered average) $\overline{s_{cur}}(\tau) = \frac{1}{T}\sum_{t=1}^{T} c(t)s(t - \tau)$, and the current-based covariance matrix was given by $C_{\tau_1,\tau_2}^{cur} = \frac{1}{T}\sum_{t=1}^{T} c(t)[s(t - \tau_1) - \overline{s_{cur}}(\tau_1)][s(t - \tau_2) - \overline{s_{cur}}(\tau_2)]$. We again subtracted an average covariance matrix (unweighted by the current) and calculated eigenvalues and eigenvectors for the result.

We generated response predictions of the current-based covariance model using the two eigenvectors with the largest magnitude, and applied the methods for fitting a two-dimensional nonlinearity described above. The performance of the resulting model is reported in *Figure 2E*, and example fits are shown in *Figure 2—figure supplement 1*. To be applied to spike trains, such methods require much more data, and we could not generate firing rate predictions of the spike-based model with reasonable accuracy given the limited data to estimate a two-dimensional nonlinearity, consistent with previous applications of spike-triggered covariance to retina data (e.g., [*Liu and Gollisch, 2015*]). Note that simply estimating separate one-dimensional nonlinearities for each filter (e.g., [*Sincich et al., 2009*]), results in significantly worse predictive performance (e.g., [*Butts et al., 2011*]), due to the non-separability of the resulting nonlinear structure, as well as the inability for such analyses to factor in spike refractoriness.

## LNK Model

We explicitly followed the methods of *Ozuysal and Baccus (2012)* in fitting the linear-nonlinear-kinetic (LNK) model to the temporally modulated spot data (*Figure 4A*, *Figure 4—figure supplement 1*). This model involves the estimation of an LN model in combination with a first-order kinetics model that governs the dynamics of signaling elements in resting state (R), active state (A) and inactivated states (I). Note that we have omitted the fourth state of the original LNK model that explains slow adaptation, because it did not improve model performance due to the small magnitude of slow adaptation we observed (*Figure 1—figure supplement 1*). The kinetics rate constants reflect how fast the signaling elements transition between states. Model parameters are fitted to the data using constrained optimization. We adapted this model to fit the spot-annulus data by extending the linear filter of the LN model into separate temporal filters for center and surround processing (*Figure 4B*). Note that parameters for more complex forms of the LNK model (e.g., those considered in *Figure 4—figure supplement 2*) cannot be tractably fit to data, and we chose the parameters of these models and simulate their output, as described in *Figure 4—figure supplement 2*.

## Statistical modeling of the spike response

We applied several statistical models to describe the spike response of ganglion cells. We first considered the generalized linear model (GLM) (*Paninski, 2004*; *Truccolo et al., 2005*). We assumed that spike responses are generated by an inhomogeneous Poisson process with an instantaneous rate. The GLM makes prediction of the instantaneous firing rate of the neuron $r(t)$ based on both the stimulus $s(t)$ and the recent history of the spike train $\mathbf{R}(t)$:

$$r(t) = F_{spk}[\mathbf{k}_{lin} \cdot \mathbf{s}(t) + \mathbf{h}_{spk} \cdot \mathbf{R}(t) - \theta], \tag{8}$$

where $\mathbf{k}_{lin}$ is a linear receptive field, $\mathbf{h}_{spk}$ is the spike-history term and $\theta$ is the spiking threshold.

Note that for fitting the model parameters of the GLM (as well as those of other models with a spike-history term $\mathbf{h}_{spk}$), we used the observed spikes for predicting firing rates (i.e., $\mathbf{R}(t)$ was

derived from the observed spike train). However, to generate cross-validated predictions, we did not use the observed spikes, and instead calculated $\mathbf{R}(t)$ iteratively as the history of a simulated spike train that was generated using a Poisson process (see *Butts et al. 2011* for more details).

The parameters of the GLM are all linear functions inside the spiking nonlinearity $F_{\mathrm{spk}}[\cdot]$. The LN model consists of only the linear receptive field and the spiking threshold; the full GLM further includes the spike-history term (denoted as LN+RP in figures). The spiking nonlinearity had a fixed functional form $F_{\mathrm{spk}}[g] = \log[1 + \exp(g)]$, satisfying conditions for efficient optimization (*Paninski, 2004*). The choice of this particular parametric form of spiking nonlinearity was verified with standard non-parametric estimation of the spiking nonlinearity (*Figure 1B,E*) (*Chichilnisky, 2001*). The model parameters were estimated using maximum likelihood optimization. The log-likelihood (*LL*) of the model parameters that predict a firing rate $r(t)$ given the observed neural response $r_{\mathrm{obs}}(t)$ is (*Paninski, 2004*):

$$LL = \sum_t [r_{\mathrm{obs}}(t) \log r(t) - r(t)] \tag{9}$$

The optimal model parameters could then be determined using gradient-descent based optimization of *LL*. Although this formulation of the model assumes probabilistic generation of spikes, this fitting procedure was able to capture the parameters of the equivalent integrate-and-fire neuron, and thus was not impacted by the form of noise related to spike generation (*Butts et al., 2011*; *Paninski et al., 2007*).

To capture nonlinear properties of the spike response, we extended the Nonlinear Input Model (NIM) (*McFarland et al., 2013*) to include multiplicative interactions. The predicted firing rate of the NIM is given as:

$$r(t) = F_{\mathrm{spk}}\left[\sum_i \zeta_i[\mathbf{s}(t)] + \mathbf{h}_{\mathrm{spk}} \cdot \mathbf{R}(t) - \theta\right], \tag{10}$$

where each $\zeta_i[\cdot]$ represents a component of [potentially nonlinear] upstream processing. The original formulation of the NIM assumed this upstream processing had the form of an LN model (making the NIM an LNLN cascade). However, based on knowledge of nonlinear processing in the synaptic current response, here we assumed that the upstream components $\zeta_i[\mathbf{s}(t)]$ take the form of a DivS model:

$$\zeta[\mathbf{s}(t)] = f_{\mathrm{e}}[\mathbf{k}_{\mathrm{e}} \cdot \mathbf{s}(t)] \times f_{\mathrm{s}}[\mathbf{k}_{\mathrm{s}} \cdot \mathbf{s}(t)]. \tag{11}$$

Similar to parameter estimation of the DivS models of synaptic current response, we alternately estimated the filters and nonlinearities until they converged. The set of constraints on excitatory and suppressive nonlinearities was the same as with DivS model of synaptic currents.

## Quantification of contrast adaptation with LN analysis

We performed a more traditional LN model analysis to gauge the adaptation to contrast of both the observed data as well as the predictions of nonlinear models, following (*Chander and Chichilnisky, 2001*; *Baccus and Meister, 2002*). We first separately performed LN analysis for each contrast level, establishing the filter by reverse correlation and then using the filter output at each contrast to separately estimate each nonlinearity. The nonlinearities were then aligned by simultaneously estimating a scaling factor for the x-axis and an offset for the y-axis to minimize the mean squared deviation between them. The associated scaling factor was incorporated into the linear filters such that contrast gain was attributable entirely to changes in the linear filter (*Chander and Chichilnisky, 2001*). While the offset parameter was not used in (*Chander and Chichilnisky, 2001*), we found it was necessary to fully describe the changes in the nonlinearities associated with the synaptic current (but not spiking) data (*Figure 1—figure supplement 1*). This additional offset is shown in the comparison between the two nonlinearities across contrast (e.g., *Figure 1E*).

Once the linear filters at both contrasts were obtained, we calculated contrast gain as the ratio of standard deviations of the filters at low and high contrast conditions. To make more detailed comparisons about the filter shape, we also calculated a biphasic index, based on the ratio of the most negative to the most positive amplitude of the LN filter $\mathbf{k}$, i.e., $|\min(\mathbf{k})/\max(\mathbf{k})|$.

## Evaluation of model performance

We fit all models on the 7 s segments of unique stimuli within each 10 s block, and cross-validated model performance on the 3 s repeat trials. We calculated the predictive power, or percent of explainable variance (*Sahani and Linden, 2003*), to quantify how well the model captured the trial-averaged response for both intracellular and extracellular recordings. This metric is based on the fraction of explained variance ($R^2$) but corrects for noise-related bias due to a limited number of trials. For validation of spike-based models, we simulated individual instances of spike trains using a non-homogeneous Poisson process, and the model predictions were based on measuring a PSTH to 500 simulated repeats in response to the cross-validation stimulus. All measures of model performance compared predicted to measured responses using 1 ms bins, which was necessary to measure how accurately the different models captured temporal precision (*Butts et al., 2011*, *2007*).

## Coherence analysis of synaptic current response

The general model performance metrics such as predictive power and cross-validated likelihood do not reveal which aspects of the response are not captured by the model. We thus devised a new coherence-based metric to quantify how well the model performs across temporal frequencies. The coherence between the model predicted current response $c(t)$ and the recorded current response on the $i$th trial $c^i_{\mathrm{obs}}(t)$ is (*Butts et al., 2007*):

$$\gamma^2_i(\omega) = \frac{\langle |\mathrm{C}^i_{\mathrm{obs}}(\omega)\overline{\mathrm{C}(\omega)}|^2\rangle}{\langle |\mathrm{C}^i_{\mathrm{obs}}(\omega)|^2\rangle \langle |\mathrm{C}(\omega)|^2\rangle} \tag{12}$$

where $\mathrm{C}(\omega)$ and $\mathrm{C}^i_{\mathrm{obs}}(\omega)$ are the Fourier transforms of $c(t)$ and $c^i_{\mathrm{obs}}(t)$ respectively, and the bar denotes complex conjugate. We used angular frequency $\omega = 2\pi f$ instead of $f$ to be consistent with common conventions. The coherence measure on individual trials was averaged across repeats for each cell.

Because the observed response on each trial contains noise, a coherence of one throughout the frequency is not a realistic target. To correct for this bias, we calculated the coherence between the trial-averaged current response (i.e., the ideal predictor of response) and the recorded current on each trial. This noise corrected coherence metric represents an upper bound of coherence that can be achieved by any stimulus-processing model. It also reflects the consistency of current response at each frequency. For example, in the low contrast condition, the response contained little high frequency components (*Figure 7A–B*), and consequently the measured coherence was close to zero above 30 Hz.

## Event analysis of spike trains

We modified a previously established method to identify short firing episodes (events) in the spike train (*Berry et al., 1997*; *Butts et al., 2010*; *Kumbhani et al., 2007*). Specifically, events were first defined in the peristimulus time histogram (PSTH) as times of firing interspersed with periods of silence lasting $\geq$ 8 ms. Each resulting event was further analyzed by fitting the PSTH with a two-component Gaussian mixture model. An event was broken into two events if the differences of means of the two Gaussian components exceed two times the sum of standard deviations. Event boundaries were defined as the midpoint between neighboring event centers and were used when assigning event labels to simulated spikes. Events were excluded from further analysis if no spike was observed on more than 50% of the trials during the event window. This criterion excluded spontaneous spikes that occurred on few trials. Event analysis was first performed on responses at high contrast. Events at low contrast were defined using the event boundaries obtained from high contrast data. These particular methods were chosen because they gave the most reasonable results with regards to visual inspection, but the results presented here do not qualitatively depend on the precise methods.

Once the events were parsed, we measured several properties associated with each event relating to their precision and reliability (*Figures 1,6*). First, we measured the jitter in the timing of the first-spike, using the SD of the first spike of the event on each trial. Then, the event time scale was estimated as the SD of all spike times in each event, which is related to the duration of each event.

Finally, the event Fano factor measured the ratio between the variance of spike count and the mean spike count in each event.

## Statistical tests

All statistical tests performed in the manuscript were non-parametric Wilcoxon signed rank tests, unless otherwise stated. All significant comparisons were also significant using t-tests.

## Acknowledgements

This work was supported by NSF IIS-1350990 (YC, DAB) and NIH EY021372 and EY014454 and an unrestricted grant from Research to Prevent Blindness to the Department of Ophthalmology & Visual Science at Yale University (YVW, SJHW, JBD).

## Additional information

### Funding

| Funder | Grant reference number | Author |
| --- | --- | --- |
| National Science Foundation | IIS-1350990 | Yuwei Cui<br>Daniel A Butts |
| National Institutes of Health | EY021372 | Yanbin V Wang<br>Silvia J H Park<br>Jonathan B Demb |
| National Institutes of Health | EY014454 | Yanbin V Wang<br>Silvia J H Park<br>Jonathan B Demb |
| Research to Prevent Blindness | | Yanbin V Wang<br>Silvia J H Park<br>Jonathan B Demb |

The funders had no role in study design, data collection and interpretation, or the decision to submit the work for publication.

### Author contributions

YC, Analysis and interpretation of data, Drafting or revising the article; YVW, Analysis and interpretation of data, Acquisition of Data; SJHP, Acquisition of data; JBD, Conception and design, Acquisition of data, Drafting or revising the article; DAB, Conception and design, Analysis and interpretation of data, Drafting or revising the article

### Author ORCIDs

Daniel A Butts, http://orcid.org/0000-0002-0158-5317

### Ethics

Animal experimentation: All experimental procedures were conducted in accordance with the recommendations in the Guide for Care and Use of Laboratory Animals of the National Institutes of Health. All of the procedures involving animals were approved by an institutional animal care and use committee (IACUC) protocol (#11431) at Yale University.

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
