## [Decision Letter]

[Editors’ note: a previous version of this study was rejected after peer review, but the authors submitted for reconsideration. The first decision letter after peer review is shown below.]

Thank you for submitting your work entitled "Divisive suppression explains high-precision firing and contrast adaptation in retinal ganglion cells" for consideration by *eLife*. Please accept our apologies for the long delay in returning these reviews to you. We had a difficult time finding qualified reviewers for your paper and then one of the initial reviewers had to bow out at a late date.

Your article has been reviewed by two peer reviewers. The process was overseen by a Reviewing Editor and by Timothy Behrens, the Senior Editor. Our decision was reached after consultation between the reviewers, the Reviewing Editor and the Senior Editor.

Based on these discussions and the individual reviews below, we regret to inform you that this submission will not be considered further for publication in *eLife*. However, the reviewers and Editors thought that your study was interesting and worthwhile, and after discussion amongst the Editors it was decided that we would be happy to consider a revised paper, which would be considered as a new submission. Based on extensive discussions with the reviewers, the Reviewing Editor believes that a resubmission would be considered if you were to revise the paper in one of three ways: (1) undertake pharmacological studies as suggested by one reviewer, and therefore try to tie the model more closely to specific neural mechanisms; (2) use a wider range of stimuli, such as stimuli that include naturalistic slow temporal fluctuations (e.g., 1/f), and then modify the model as necessary to account for a broader range of phenomena; (3) try to get a better sense of which elements of the model are critical under what situations, but testing an even wider range of competing models and model variants. We thank you for sending your work for review and we hope you will either revise this work and resubmit it, or barring that you will wish to submit to *eLife* again in the future.

Reviewer #1:

This paper studies the responses of mouse α retinal ganglion cells responding to flickering white noise at two different contrast, and fits the response with a computational model. The model is an abstract model consisting of two pathways, with one influencing the other, which is one of a general class of known models with multiple pathways followed by a nonlinear static (time-independent) interaction. It is shown that the model accurately captures the response and some properties of contrast adaptation. An argument is made why the model can be interpreted to show that inhibition underlies contrast adaptation. Other aspects of the model such as the source of precise firing are discussed. Currently the claim of the role of inhibition is not well supported, and the result that a more simple model of a known type fits the data for one type of ganglion cell is interesting, but mainly to the community of people that fit models of the retina.

Major points

1) A central claim of the paper is that the model indicates that adaptation comes from inhibitory suppression. This claim is in conflict with other studies, including one from the Demb lab (Brown & Masland, Manookin & Demb 2006). The current evidence for this claim is not sufficient, and a pharmacological experiment blocking inhibition has to be done for this claim to be supported. If it turns out that blocking inhibition abolishes adaptation and changes the response of the model in the expected way, then this should be the main focus of the paper, and the model should be used to support this main claim. In doing this experiment, the primary concern will be that without inhibition, the retina is put in a different state, where adaptation does not occur, even though inhibition is not really involved. For example, without inhibition, neurons will depolarizes and synapses may be depressed even at low contrast. So care must be taken to avoid this issue. In particular, the expectation is that the excitatory direction changes little, but the suppressive direction disappears. Because the low contrast is 10% (which is still fairly strong) it may be useful to use a lower contrast.

2) The authors interpret the 'suppressive' direction to mean inhibition, but the alternative interpretation that the authors wish to argue against is that feedback at excitatory synapses produces an apparent suppression when fit with the DivS model. It is quite suspicious that the suppressive direction (temporal filter) in Figure 2 looks exactly like a delayed version of the excitatory temporal filter. This seems to imply that the suppressive direction might indeed be from negative feedback – though admittedly such negative feedback could from synaptic depression or inhibition. The results presented are consistent with the current model of synaptic depression (not inhibition) underlying adaptation, except for Figure 4, which relies on a questionable assumption (discussed below). That's why the pharmacological test is critical.

3) As the sole support of the claim that the suppressive component of the model comes from inhibition, a spatial stimulus is used in Figure 4. This brings a new level of complexity is not properly addressed. When space is considered, we know that bipolar cells have a receptive field center and surround with different temporal filters. Bipolar cells form nonlinear subunits, each with a threshold (Schwartz et al., 2012 for mouse α cells). Each of these small subunits show local adaptation (Brown & Masland, 2001) and then there is global adaptation in the ganglion cell from spike generation (Zaghloul et al. 2005). The logic the authors use is the suppressive direction should be the same as the excitatory direction if inhibition is not involved. It is not clear whether this is true when the more complex system is considered. The 'excitatory' and 'suppressive' directions are just the two directions in stimulus space that influence the response, and the suppressive direction will be influenced by feedback at bipolar cell synapses and in the ganglion cells. The surround time course is more delayed, and may match the delayed suppression from negative feedback more closely. Thus it is not clear whether the simple DivS model will mix inputs in unexpected ways. The fact that the suppressive and excitatory directions are different is not conclusive without further consideration of the known sites of nonlinearities and adaptation.

4) Because the model has no slow dynamics, it cannot capture contrast adaptation that lasts over several seconds, which has been the sole subject of some studies of contrast adaptation (Manookin & Demb, 2006). The entire 20 s stretch of recording should be shown of the data, so it can be properly evaluated what aspects of the response cannot be fit by the model. Furthermore, the claims about the accuracy of the model should be qualified in terms of the aspects of adaptation that it does not capture.

5) Class of model. If in fact there are only two stimulus directions that are important as is claimed, than the DivS model is equivalent to a model that can be created using standard Spike-Triggered Covariance (STC) analysis (Fairhall et al., 2006), which can equivalently be done with continuous currents. So it should be pointed out that this model is not a new type of model, but an example of a known type of model. In (Fairhall, 2006) there are multiple types of ganglion cells reported with different types of responses in the two-D feature space, yet contrast adaptation is widespread among ganglion cells, so it is likely that the interpretations of the current paper won't apply generally. It is ok to study one type of cell, but the fact that the model presented is one of a known class of models should be made clear.

6) The LNK model is not convex, and more detail should be given as to how this was optimized. Were the methods of (Ozuysal & Baccus, 2012) used? For example, mean squared error was not used as an error metric. If the method of fitting was suboptimal, then the interpretation of the comparison between models is not clear. Also, more detail needs to be given for the spatial LNK model. Because there are two different spatial regions, each one should have independent adaptation, and each spatial region should have a center-surround linear receptive field. This would be a new type of model, so it's not clear whether this model can be properly optimized.

7) For the spiking DivS model, does the model really take the observed spike train as an input (not just as a constraint to fit data)? Shoudn't a model that takes the response as an input be able to produce the response? (It seems like cheating.) In fact, some studies that use the spike history in the way have shown that using the spike history is either the most important predictor of the response, or that the spike history alone can be used to predict the response (Kraus et al., 2015, Figure S4; Trucculo, et al., 2010). If this is really what was done, this approach differs from other models which use a spike-history term such as a generalized linear model (GLM) (Pillow et al., 2008) but that generate its own spikes rather than taking the actual response as an input. This needs to be made more clear, and also justified as to why this is useful. This approach may be useful for understanding the source of precision as the authors do, but it doesn't seem impressive that this model can fit the data if it really takes the actual response as an input.

8) Figure 6, it is not clear where the noise comes from in the model, as there is no noise source. Is it all from the spike-history term, i.e. the actual data? It's not clear how we can evaluate different models using this approach, because the noise is not appropriate for each model. A better approach would be to model noise sources and put them in each different model, like an LNP model.

Reviewer #2:

This paper shows how a novel model can predict the retinal ganglion cell responses to full field flicker at different contrasts. Standard approaches, like LN models, fail to generalize across contrasts, while this model does generalize with a single set of parameters to predict responses to high and low contrasts.

The key novelty of this model is to include both excitatory and inhibitory subunits, that are multiplied together to predict the ganglion cell membrane potential. Further non-linear processing allows to predict the spiking response. The model has been carefully fitted to both intracellular and extracellular recordings. The agreement between the model prediction and the data is impressive.

Overall this is a very nice work. Contrast adaptation is ubiquitous in the visual system. Such a model could have a broad impact on the field of sensory systems.

Major comment:

My only concern is about novelty: as noticed by the authors, another model, the LNK by Ozuysal and Baccus, has already shown its ability to predict ganglion cell responses at different contrasts. The performance of the current model is better than the LNK, but not by much for full field stimuli. The only stimulus where there is a significant difference, if I understood well, was the one composed of a center spot and an annulus stimulating the surround. There the model performed better, but the difference is rather small (around 7%).

The model is conceptually novel, but relatively similar to previous works by Sahani and colleagues.

So I think a more thorough comparison between the LNK model and this one would be welcomed. With such a small difference, one would like to be sure that the authors gave the LNK model its best chance of success. Here the methods do not include a text about how this model was fitted. More generally, more details would be welcomed about how the two models were fitted in the case of the compound stimulus (centered spot +annulus).

[Editors’ note: what now follows is the decision letter after the authors submitted for further consideration.]

Thank you for resubmitting your work entitled "Divisive suppression explains high-precision firing and contrast adaptation in retinal ganglion cells" for further consideration at *eLife*. Your revised article has been favorably evaluated by Timothy Behrens (Senior editor), a Reviewing editor, and two reviewers.

The manuscript has been improved but there are some remaining issues that need to be addressed before acceptance. As you will see Reviewer 1 was very positive, but Reviewer 2 had several remaining concerns. In an extensive consultation session between both reviewers and the Reviewing Editor, it was decided that the best course of action would be for us to return this to you for further revision to meet the concerns of Reviewer 2.

1) The biggest remaining issue that must be addressed in revision concerns issue #1 raised by Reviewer #2, the use of raw data in prediction. This issue was also brought up in the initial reviews. All the discussants are concerned there is a serious danger of over-fitting here, and they did not consider the responses to the initial reviews on this matter to be sufficient. Reviewer 1 makes some good suggestions about how to deal with this issue, but you might also be able to approach it another way.

2) Please try to address the slow contrast adaptation issue as suggested by Reviewer 1. If it cannot be addressed then the manuscript should be revised to make it clear that this mechanism is not covered by the model.

3) Please respond to the dynamics/LNK issue with appropriate revisions (preferably including the requested comparisons).

4) The presentation regarding the Spike Triggered Covariance model and its relationship to the DivS model needs to be clarified.

5) Please revise the text concerning divisive suppression to address Reviewer #1's concerns.

As the discussants see it, you can meet these concerns in one of two ways:

Option 1: Revise and retract some of the claims so that the claims are more consistent with what is actually shown in the existing manuscript.

Option 2: Run the model again, but using a spike history filter generated from synthetic spikes rather than using the raw data in the model prediction as was done in the most recent revision.

The discussants agreed that either of these two options would be fine and that they would leave it up to you to decide which course of action you would like to pursue.

In either case we appreciate your submitting this excellent manuscript to *eLife* and we hope that these remaining revisions will not be too onerous for you to accomplish quickly.

Reviewer #1:

As far as I am concerned the authors replied well to the points I raised. It is clear that this study shows a novel model that can predict better ganglion cells responses. The quantitative improvement compared to previous approaches can be small, but the framework seems a bit more general. It is also stated clearly that this model by itself does not lead to novel insights about the mechanisms behind contrast adaptation as different mechanisms can explain these results.

Reviewer #2:

The paper has improved from the previous version, but a number of issues remain that were raised in the first review. The main positive points are that it captures contrast adaptation in a particular cell type with a simple model, and that it raises a new potential mechanism for adaptation using one pathway modulating another, which is different from the accepted explanation of synaptic depression. It can be made suitable for publication, but a number of things must be done.

1) Spike history. For the spiking DivS model, the authors use the raw data as an input to the model when predicting the data. This seemed to be the case in the last review, although it was baffling to me and surprised both reviewers. Insufficient justification has been given for this procedure, and I can't conceive of any justification for it. The arguments given by the authors for this procedure are not persuasive, and as stated in the last review, this seems like cheating to get the model to predict more accurately.

Some arguments given by the authors as to why it is ok to use the data to predict the data:

“First, as detailed in the methods, we constrain the spike history term to be negative, meaning it cannot be used to predict future spikes, but rather only influences temporal patterning on short time scales due to refractoriness.”

If the spike history term is negative, thus causing silences, it helps the model predict future silences, which the same thing as changing the prediction of future spikes.

And Figure 5 shows the authors' claim that it cannot be used to predict future spikes isn't correct. With the spike history coming from the data, the beginning of bursts become more sharp, meaning the spike history is long enough for the data spikes in one burst to influence onset of spiking in the model's next burst. The decreased jitter in the statistics for the first spike in a burst confirms this.

Furthermore, the spike history goes out at least to 40 ms, and the full duration isn't shown or stated. The fact that it is small at 40 ms doesn't matter, because at long timescales the integration over multiple spikes will produce a large cumulative effect.

“Second, as we demonstrate, using a spike history term with an LN model (the LN+RP model in later figures) has poor performance in high contrast (Figure 5), and also cannot explain contrast adaptation effects (Figure 7), and thus the spike-history term is only effective at explaining these effects in tandem with divisive suppression in a two-stage computation, which is a major point of this work. “

This just means that this inappropriate use of the data in the model is not sufficient alone to predict the response, but it is necessary. It is nonetheless using the data to predict the data.

“In these figures, we explicitly compare the spiking DivS model with the LN+RP model (without DivS) to show that DivS is necessary, and furthermore show that the DivS-RP model (without spike history) also cannot explain the response to high precision and contrast adaptation, demonstrating that it is the interplay of spike-history effects with the DivS computation that uniquely explain the ganglion cell response and its adaptation to contrast. As described above, the use of spike history is well vetted by much published work in modeling the retina, but the DivS model goes well beyond this to show it alone is not sufficient to explain ganglion cell firing, and must be combined with computations present at the input to ganglion cells. “

The use of spike history in a feedback loop as part of the model is acceptable of course, but using the data spikes in the model's prediction is not. It is not well vetted to use this procedure to draw conclusions about the model performance, and as mentioned in the previous review references exist to show that it can greatly improve a model prediction (Kraus et al., 2015, Figure S4; Trucculo, et al., 2010), which shows why it is not an acceptable procedure.

The following statements and conclusions are currently unsupported, and need to be supported by a model using a spike history feedback filter, and not spike history from the raw data. It's fine to take these conclusions about millisecond precision and full adaptation in the spiking model out of the paper, the paper could stand without them.

Abstract. "The full model accurately predicted spike responses with unprecedented millisecond precision, and accurately described contrast adaption of the spike train."

Introduction. "Ganglion cell firing, further shaped by spike generation mechanisms, could be predicted to millisecond precision."

Figure 5–Figure 6 indicate that without the raw data (Div – RP model), predictions are no better than an LN model. Claims about precision should be removed unless a full spiking model with feedback spike history (without the raw data) is implemented.

Subsection “Contrast adaptation relies on both divisive suppression and spike refractoriness” "Therefore, the two nonlinear properties of retinal processing, contrast adaptation and temporal precision, are tightly related mechanistically and can be simultaneously explained by the divisive suppression model."

Contrast adaptation in spiking for the Div – RP model is improved over an LN model, so that claim can be kept, although full adaptation is not captured.

2) Lack of slow contrast adaptation. There is currently insufficient evidence addressing the lack of slow contrast adaptation, and in fact the authors perform an analysis that would obscure evidence of slow adaptation. Unfortunately, the authors have resisted my request to show the raw data from a 20 s segment of the recording, which would show the change to both high and low contrast and the full 10 s recording at each contrast. It's ok if there is some slow adaptation, but they have to show whether it's there or not.

Slow adaptation is revealed only in the change in the average membrane potential (Manookin & Demb, 2006). Gain and temporal filtering do not adapt slowly (Baccus & Meister, 2003), but statistics for gain and temporal filtering are the focus of the supplemental figure. The only evidence that might have been in the figure is the vertical position of the nonlinearity, which would show the change in average membrane potential. But surprisingly, in the methods it states, citing (Chander & Chichilnisky, 2001) "The resulting nonlinearities were then aligned by introducing a scaling factor for the x-axis and an offset for the y-axis." This y-axis offset would remove the evidence of slow adaptation. This offset was not used in (Chander & Chichilnisky, 2001) and should not be used here.

To be clear, three things must be done to address whether there is slow adaptation. 1) A full raw trace must be shown. 2) Nonlinearities must be shown without manipulation of the y-axis offset. 3) Because (Manookin & Demb, 2006) shows that most slow adaptation decays by 1.5 s, they should show the membrane potential averaged over trials and binned in increments of no larger than 1 s across the full 20 s.

3) LNK model. The authors are trying to claim that a model of synaptic depression can't reproduce the effect that there is greater suppression is in one spatial location than another. It is good that the authors have tried to compare the DivS model with different types of LNK models having two pathways, but in order to rule out a possibility (localized suppression from a depression model), they have to show that they have sufficiently tried to make that possibility work.

The authors resisted my previous suggestion in the first round of review to consider that bipolar cells have center-surround receptive fields and that more than one bipolar cell feeds into a ganglion cell. This implies that for two pathways, one from the center and one from an annulus, each tested pathway should have an input from both central and peripheral regions. The DivS models was allowed to have two pathways, each with a different weighting from the center and surround, but the LNK model does not have such a mixture for each pathway. A model should be tested where each of the two pathways of the LNK model have a weighting of center and surround. To be clear, one pathway should represent bipolar cells in the center, and should have a stronger weighting from the center than the annulus. The other pathway should represent bipolar cells whose receptive field center is under the annulus region, and should have a stronger weighting from the annulus than the central spot. This fits most with known circuitry, because bipolar cells have center-surround linear receptive fields. This test will probably work as the authors predict, but the current comparisons are not adequate for their claim.

The previously published LNK model used a 4-state kinetic model, the current version of the paper now shows that a 3-state model was used here. This means that the simpler model chosen here can't reproduce all of the dynamics of the previous model. This difference needs to be clearly pointed out and justified in the main text. This issue also fits with showing whether that there is not slow adaptation, because it may be that the reason that the LNK model slightly underperforms the DivS model is that they chose a model with more simplified dynamics, yet the cell has both fast and slow dynamics.

4) Comparison to Spike Triggered Covariance. There is some confusion in the presentation about what STC analysis reveals, and the relationship to the DivS model.

“We added Figure 2—figure supplement 1 to fully describe an STC analysis and added modeling data to Figure 2. As explained >above, we more clearly demonstrate a number of key differences of the DivS >model over the STC model, most notably the presence of divisive suppression. >We now make clear the caveat that other cell types could require additional >components to capture important features of the response in the Discussion.”

The following is a summary of the relationship, and it is basically in agreement with what is stated in the legend of Figure 2—figure supplement 1: STC analysis produces orthogonal vectors that define the stimulus subspace that drives the cell's response. It does not speak to the subsequent nonlinear mapping of that subspace to the response. In this case, if there is a two-dimensional subspace that defines the response, the filters of the DivS models must occupy the same subspace two STC eigenvectors. Thus this subspace can be found by a standard STC analysis not requiring a optimization procedure.

For the choice of nonlinear mapping of the subspace to the response, either there can be an n-D nonlinearity (2-D in this case), which would be the optimal solution, or a more simplified nonlinearity can be found. This is what the DivS model does, to simplify the 2-D nonlinearity to a product of 1-D nonlinear functions.

The reason I requested this detailed comparison was simply to avoid the mistaken conclusion that the DivS model is really a new class of model. It finds a reduced dimensional subspace just as STC does, and in fact such a standard technique can be used. The DivS model then does provide a set of constraints to reduce the complexity of the subsequent nonlinearity, reducing a 2-D nonlinearity to 1-D functions.

Even though the Figure 2—figure supplement 1 legend basically agrees with these statements, there are several statements and analyses in the paper that indicate a confusion:

Subsection “The nonlinear computation underlying synaptic inputs to ganglion cells”. However, the 2-D mapping between STC filter output and the synaptic current differed substantially from the same mapping for the DivS model (Figure 2—figure supplement 1).

This isn't possible if things were done correctly (and it seems to conflict with the Figure 2—figure supplement 1 legend), the STC subspace should be the same as the DivS subspace, and the 2-D nonlinearity from the STC subspace should be the same as the mapping from subspace to response for the DivS model. I think this is just a problem of how something is being stated, but it should be corrected/clarified.

There is one potential difference between a full-2D nonlinearity from an STC subspace and the DivS model, and that is that the full 2-D nonlinearity may overfit, and thus the DivS model may impose a regularization that allows a better model of a test data set. If this is true, this is an interesting point to make, and it seems to be more relevant with the spiking model, because the authors say that can't fit the 2-D nonlinearity for spiking data (they can of course, but it must not be accurate).

In the same section, Thus, despite the ability of covariance analysis to nearly match the DivS model in > terms of model performance (Figure 2), it could not uncover the divisive interaction between > excitation and suppression (Figure 2).

That's not the point of STC analysis, it only acts to find the relevant subspace. A second step would define the nonlinear mapping from subspace to response, either preserving the full 2-D nonlinearity, which would be the optimal solution, or simplifying the 2-D nonlinearity to a more biological combination of 1-D pathways.

It seems what the analysis of Figure 2 has done was to use the eigenvectors as filters for subsequent 1-D nonlinearities. That doesn't make any sense, the ideal 1-D nonlinearities have to lie in the 2-D subspace, but the 1-D nonlinearities don't have to be the eigenvectors, or even be orthogonal. This is not part of a standard STC analysis, and there is no reason to suspect that this would work.

The point of my request to state the relationship between the DivS model and STC analysis was not to pit STC analysis against the DivS model, because they should yield equivalent results as the stimulus subspace, and that's as far as STC analysis goes. It was so say that STC analysis will find the equivalent subspace as the DivS model, and that the DivS model provides a way to simplify the nonlinear mapping.

Discussion section, paragraph two. The presence of nonlinearities that are fit to data in the context of multiplicative >interactions distinguishes this model from multi-linear models (two linear terms multiplying) >(Ahrens et al., 2008a; Williamson et al., 2016), as well as more generalized LN models such as >those associated with spike- triggered covariance.

This isn't clear, STC only identifies the subspace, and then multiplicative interactions can be identified within that subspace.

5) Circuit mechanisms underlying divisive suppression.

It is appropriate to discuss the possibility that inhibition does play a role in adaptation, but the authors have to mention the previous literature that says that inhibition isn't needed for contrast adaptation (Brown & Masland, 2001; Manookin & Demb, 2006). This is in their favor to do so, because they point out that their model suggests an unexpected mechanism for contrast adaptation.

There is another point that the author's may wish to mention that supports their argument. As they mention, On-α cells are more linear with a high spontaneous firing rate. Depression models rely on there being a change in the average level of synapse activation with contrast. With a linear cell, the average input to the synapse doesn't change much with contrast, and thus a change in the level of depression may not occur. Thus, inhibition from a modulatory pathway may be needed to create contrast adaptation for a more linear cell.

6) Meaning of Divisive Suppression. The authors use the term divisive suppression to apply both to one pathway modulating another (the usual term), and to any change in gain as might occur from synaptic depression. This second use is strange, if that's true, then the finding of divisive suppression isn't new, they have just called gain control something else.

The potential new concept is that the effect is better modeled by one pathway modulating the other. This is actually an old concept, (Victor) and has since been replaced by one where a second pathway is not necessary, namely synaptic depression. But the new result would be that the older concept of modulation works better, especially when one considers the spatial stimulus. To make this comparison and state this conclusion, it makes more sense to restrict 'divisive suppression' to mean modulation, and thus the question becomes one of Suppression vs. Depression, rather than saying depression is one type of suppression.

---

## [Author Response]

[Editors’ note: the author responses to the first round of peer review follow.]

*Based on these discussions and the individual reviews below, we regret to inform you that this submission will not be considered further for publication in eLife. However, the reviewers and Editors thought that your study was interesting and worthwhile, and after discussion amongst the Editors it was decided that we would be happy to consider a revised paper, which would be considered as a new submission. Based on extensive discussions with the reviewers, the Reviewing Editor believes that a resubmission would be considered if you were to revise the paper in one of three ways: (1) undertake pharmacological studies as suggested by one reviewer, and therefore try to tie the model more closely to specific neural mechanisms; (2) use a wider range of stimuli, such as stimuli that include naturalistic slow temporal fluctuations (e.g., 1/f), and then modify the model as necessary to account for a broader range of phenomena; (3) try to get a better sense of which elements of the model are critical under what situations, but testing an even wider range of competing models and model variants. We thank you for sending your work for review and we hope you will either revise this work and resubmit it, or barring that you will wish to submit to eLife again in the future.*

We appreciate the opportunity to resubmit our manuscript to *eLife*. According to our previous decision letter, the Editors recommended choosing one of three directions to expand our study: (1) pharmacology; (2) additional stimuli; (3) computation. We chose option 3: to test a wider range of competing computational models and model variants to clarify which elements of our DivS model are critical for capturing ganglion cell responses, with high temporal precision, at the levels of both synaptic currents and spikes.

A) The main advance of our paper is the identification of divisive suppression as a critical computation in retinal ganglion cells. We discovered a computational architecture that accurately captures the response properties in synaptic current. We expanded this description to capture spikes at millisecond resolution. In the process, we show that computational mechanisms identified in the synaptic current inputs contribute to high temporal precision in the spikes as well as to contrast adaptation. In the revision of the manuscript, we try to make clear the computational advance in our study and how our approach could inspire similar analyses in other parts of the brain.

B) We chose not to expand our study with additional pharmacological experiments at this time (editors’ option 1, above). Our Figure 4 shows that a component of divisive suppression primarily originates in surround regions of the receptive field, whereas excitation primarily originates within the center region. The incomplete overlap between excitation and suppression is a novel finding, which we clarified through further material and revisions (see point E below). However, we realize that this observation does not rule out a combination of mechanisms for suppression, including both synaptic depression and a circuit mechanism for lateral inhibition. We now make this clear in the manuscript. We have a longer-term interest in delving further into the synaptic mechanism, but have chosen to keep the focus in this manuscript on the computation. Pharmacological experiments are not straightforward, because (i) blocking all [GABAergic + glycinergic] inhibition results in severe changes in mouse retinal function (Toychiev et al., 2013; our unpublished observations) and (ii) blocking receptors can alter the release of glutamate, which can in turn affect synaptic depression as well other effects on network function. We do hope the modeling approach presented in this manuscript can be used in tandem with such pharmacology to analyze underlying synaptic mechanisms in future studies. In this manuscript, we have therefore focused on the computational description of suppression and tempered the conclusions regarding specific synaptic mechanisms.

C) We added “covariance” analysis similar to spike-triggered covariance (STC) that can be applied to synaptic currents (i.e., continuous, event-free data). Further, we derived methods to make predictions with this analysis (using a 2-D nonlinearity that can be fit to data as part of the covariance model) – not commonly done with STC – allowing for direct comparisons to other models considered in the manuscript. Our analysis shows that covariance analysis can identify the feature space that drives the ganglion cell response (consistent with previous STC analyses of spikes), but it does not identify the critical divisive interaction identified by the DivS model. Likewise, while the covariance model coupled with a 2-D nonlinearity can generate nearly as good predictions of synaptic currents, the number of parameters it requires, as well as its inability to fit simultaneously with spike-refractoriness, make it unable to be used for accurate predictions of spikes. By comparison our spike responses (~1 minute of unique data at each contrast) could routinely be fit by the DivS model, yielding unprecedented accuracy of any spike-based model of sensory neurons. We believe that our new analysis (Figure 2—figure supplement 1) makes a valid and important comparison between STC and DivS models, and shows the advantages of the DivS model. This is coupled with comparisons in the revised manuscript to other modeling approaches, underscoring the uniqueness of the DivS model and its ability to identify the computations performed by the retinal circuit.

D) We have analyzed the stability of responses during each contrast half-cycle and examined possible effects of slow adaptation (Figure 1—figure supplement 1). Our findings clearly show (i) highly stable responses and (ii) no sign of slow contrast adaptation. Notably, an earlier study of slow contrast adaptation (Manookin and Demb, 2006) focused on OFF Α ganglion cells in guinea pig, whereas here we focus on ON Α ganglion cells in mouse. We would have to modify the DivS model to account for slow contrast adaptation in other cell types in the future: such slow mechanisms likely act in tandem with those identified in this study.

E) We demonstrate that the DivS model captures the form of synaptic depression implemented in the standard version of the LNK model (Figure 4—figure supplement 1). We have expanded this analysis to look at more complex forms of the LNK model that include both center and surround components (Figure 4—figure supplement 2). There are many possible variations of the LNK model, and we considered two reasonable cases where nonlinear center and surround components sum before applying synaptic depression vs. separate center and surround LNK models sum following depression. In both cases, we find that the DivS model can capture the effects of depression, and thus encompass mechanisms described by the LNK model. Furthermore, we show that in all such models, the excitation and suppression are always matched in their relative strengths in center vs. surround; whereas the recorded ganglion cell responses (Figure 4) analyzed with the DivS model show dissociation between strong excitation in the center and relatively strong suppression in the surround.

Collectively, the data and analysis in Figure 4 and the new analysis of the LNK model validate our conclusion that suppression has a wider spatial footprint than excitation. This pattern is consistent with a component of suppression that originates in a circuit mechanism (and is inconsistent with the LNK model), demonstrating the novelty of this result. As noted above, however, we cannot rule out a contribution from synaptic depression over the central region, and have clarified this point in the revised manuscript.

*Reviewer #1:*

*This paper studies the responses of mouse α retinal ganglion cells responding to flickering white noise at two different contrast, and fits the response with a computational model. The model is an abstract model consisting of two pathways, with one influencing the other, which is one of a general class of known models with multiple pathways followed by a nonlinear static (time-independent) interaction. It is shown that the model accurately captures the response and some properties of contrast adaptation. An argument is made why the model can be interpreted to show that inhibition underlies contrast adaptation. Other aspects of the model such as the source of precise firing are discussed. Currently the claim of the role of inhibition is not well supported, and the result that a more simple model of a known type fits the data for one type of ganglion cell is interesting, but mainly to the community of people that fit models of the retina.*

We appreciate these thoughtful comments. As noted above (see Author Points A, B, and E), we hope that we now make the point more clearly that our study’s main achievement is demonstrating how “divisive suppression” is a crucial component of retinal computation, and how it is part of a two-stage computation, which altogether can almost perfectly explain ON-Alpha responses — both their high temporal precision, and their adaptation to contrast. We now give a more balanced description of the underlying mechanism for divisive suppression, which is probably a combination of a circuit mechanism and synaptic depression. However, the precise mechanistic source of divisive suppression is not the main purpose of this work, and most of the novel results relate to identifying computations in the synaptic input, compared with those within the ganglion cell spike output, and demonstrate how precision and contrast adaptation might be simply explained in this description.

We have taken the reviewer’s comments to heart in this light, but rather than addressing the mechanism with more targeted experiments — which is not central to our main results — we instead rewrote many sections of the manuscript (as described below) to emphasize the achievements of this work more clearly. Furthermore, we added new modeling results to show that the divisive mechanism could explicitly capture synaptic depression, and thus offers a broader framework with which to consider ganglion cell computation (and the means to characterize it).

*Major points*

*1) A central claim of the paper is that the model indicates that adaptation comes from inhibitory suppression. This claim is in conflict with other studies, including one from the Demb lab (Brown & Masland, Manookin & Demb 2006).*

See Author points B and D above; we give a more balanced description of the mechanism. Furthermore, we now demonstrate a lack of slow adaptation in our recordings of mouse ON-Alpha cells (Figure 1—figure supplement 1).

*The current evidence for this claim is not sufficient, and a pharmacological experiment blocking inhibition has to be done for this claim to be supported. If it turns out that blocking inhibition abolishes adaptation and changes the response of the model in the expected way, then this should be the main focus of the paper, and the model should be used to support this main claim. In doing this experiment, the primary concern will be that without inhibition, the retina is put in a different state, where adaptation does not occur, even though inhibition is not really involved. For example, without inhibition, neurons will depolarizes and synapses may be depressed even at low contrast. So care must be taken to avoid this issue. In particular, the expectation is that the excitatory direction changes little, but the suppressive direction disappears. Because the low contrast is 10% (which is still fairly strong) it may be useful to use a lower contrast.*

See Author point B above. We agree that this experiment could be complicated and feel that our paper should remain focused on the advances in the computational description. We now give a more balanced description of likely mechanisms for divisive suppression, which does not rule out a contribution from synaptic depression.

*2) The authors interpret the 'suppressive' direction to mean inhibition, but the alternative interpretation that the authors wish to argue against is that feedback at excitatory synapses produces an apparent suppression when fit with the DivS model. It is quite suspicious that the suppressive direction (temporal filter) in Figure 2 looks exactly like a delayed version of the excitatory temporal filter. This seems to imply that the suppressive direction might indeed be from negative feedback – though admittedly such negative feedback could from synaptic depression or inhibition. The results presented are consistent with the current model of synaptic depression (not inhibition) underlying adaptation, except for Figure 4, which relies on a questionable assumption (discussed below). That's why the pharmacological test is critical.*

See Author points B and E above. We now show that synaptic depression, as implemented by the LNK model, does indeed yield the delayed suppressive filter that otherwise resembles the excitation filter (Figure 4—figure supplement 1 and Figure 4—figure supplement 2). However, the result from Figure 4 shows that excitation and suppression do not have the same spatial footprint, which cannot be explained by the mechanism implemented by the LNK model. This provides evidence for a contribution from an alternate circuit mechanism to divisive suppression, although we cannot rule out additional contributions from synaptic depression to stimuli presented in the center (i.e., that overlaps the region of excitation). We now make this clear in the manuscript.

*3) As the sole support of the claim that the suppressive component of the model comes from inhibition, a spatial stimulus is used in Figure 4. This brings a new level of complexity is not properly addressed. When space is considered, we know that bipolar cells have a receptive field center and surround with different temporal filters. Bipolar cells form nonlinear subunits, each with a threshold (Schwartz et al., 2012 for mouse α cells). Each of these small subunits show local adaptation (Brown & Masland, 2001) and then there is global adaptation in the ganglion cell from spike generation (Zaghloul et al. 2005). The logic the authors use is the suppressive direction should be the same as the excitatory direction if inhibition is not involved. It is not clear whether this is true when the more complex system is considered. The 'excitatory' and 'suppressive' directions are just the two directions in stimulus space that influence the response, and the suppressive direction will be influenced by feedback at bipolar cell synapses and in the ganglion cells. The surround time course is more delayed, and may match the delayed suppression from negative feedback more closely. Thus it is not clear whether the simple DivS model will mix inputs in unexpected ways. The fact that the suppressive and excitatory directions are different is not conclusive without further consideration of the known sites of nonlinearities and adaptation.*

The complexity of the retinal circuit when spatial stimuli are involved was the main motivation for the design of the stimulus in Figure 4, which only introduces two spatial components (center and annulus) to limit the number of possible interactions between nonlinear components. We believe this experiment (and resulting analysis) is novel both because (1) the experimental data can distinguish between our model of divisive suppression and *existing* models of ganglion cell processing (including the LNK model of synaptic depression) that can be fit to data, and (2) it reveals that new mechanisms must be present, in the sense that existing models cannot explain the data. Furthermore, the DivS model can explain the response of this more challenging stimulus nearly perfectly (>90% explainable variance at millisecond resolution) without needing to consider all these additional complexities, suggesting the simple divisive mechanism presented in this manuscript is sufficient to capture such complexities. To our knowledge this level of precision has not even been tested in any other study to date, much less reproduced using a simple two-pathway model.

Furthermore, we have now performed a number of additional simulations (Figure 4—figure supplement 2) to investigate several more complex models involving rectification and the application of synaptic depression in various combinations. The conclusions of this are two-fold: (1) in all these simulations, the source of divisive suppression still strongly correlates with that of excitation, as identified by the DivS model, which is not the case for the observed data; (2) the DivS model can still predict the response as generated by these more complex mechanisms relatively accurately, although it describes the observed ganglion cell responses with much better performance compared with data generated through these more complex models. Other modeling approaches, including the LNK model as well as covariance-based models, cannot currently scale to capture the multiple interactions in this complex model, due to the lack of convexity (which the reviewer refers to below in point 6 for the LNK model) and the dependence on response-weighted averaging for covariance-based models. Thus, these supplemental figures also demonstrate the unique ability of the DivS model to characterize these more complex interactions.

We recognize that we cannot eliminate all alternative explanations, and have clarified the purpose of the experiment and analyses considered in Figure 4 (see Author Points B and E above). We hope to have clarified that this material is useful for distinguishing between different models, and clarified the general utility of the DivS model in characterizing retinal computation, regardless of mechanism.

*4) Because the model has no slow dynamics, it cannot capture contrast adaptation that lasts over several seconds, which has been the sole subject of some studies of contrast adaptation (Manookin & Demb, 2006). The entire 20 s stretch of recording should be shown of the data, so it can be properly evaluated what aspects of the response cannot be fit by the model. Furthermore, the claims about the accuracy of the model should be qualified in terms of the aspects of adaptation that it does not capture.*

The sections of the data and model predictions that currently are shown (Figure 3, Figure 4, Figure 6, Figure 7) are zoomed to demonstrate the high-temporal resolution that the model fits, and we have thus added measurements demonstrating the lack of slow dynamics in the recordings (Figure 1—figure supplement 1; see Author Point D above). Furthermore, we now mention several components of adaptation present in other ganglion cells that could be addressed in future work, in the last paragraph of the Discussion.

*5) Class of model. If in fact there are only two stimulus directions that are important as is claimed, than the DivS model is equivalent to a model that can be created using standard Spike-Triggered Covariance (STC) analysis (Fairhall et al., 2006), which can equivalently be done with continuous currents. So it should be pointed out that this model is not a new type of model, but an example of a known type of model. In (Fairhall, 2006) there are multiple types of ganglion cells reported with different types of responses in the two-D feature space, yet contrast adaptation is widespread among ganglion cells, so it is likely that the interpretations of the current paper won't apply generally. It is ok to study one type of cell, but the fact that the model presented is one of a known class of models should be made clear.*

See Author point C. above. We added Figure 2—figure supplement 1 to fully describe an STC analysis and added modeling data to Figure 2. As explained above, we more clearly demonstrate a number of key differences of the DivS model over the STC model, most notably the presence of divisive suppression. We now make clear the caveat that other cell types could require additional components to capture important features of the response in the Discussion

*6) The LNK model is not convex, and more detail should be given as to how this was optimized. Were the methods of (Ozuysal & Baccus, 2012) used? For example, mean squared error was not used as an error metric. If the method of fitting was suboptimal, then the interpretation of the comparison between models is not clear. Also, more detail needs to be given for the spatial LNK model. Because there are two different spatial regions, each one should have independent adaptation, and each spatial region should have a center-surround linear receptive field. This would be a new type of model, so it's not clear whether this model can be properly optimized.*

We apologize for omitting in the previous manuscript how the LNK model was fit, and now include an additional section of the Methods describing this (lines 559-568). In brief, we explicitly followed the methods of Ozuysal & Baccus, although we needed to use different model initializations due to the much faster time courses of the neurons in our study. When fit to temporally modulated spot data, the LNK model is quite successful (Figure 4), validating our fitting approach. We extended the LNK model to the center-annulus stimulus (as now described in the Methods), and have added additional detail about this extension in added figure panels (Figure 4), as well as in two supplemental figures (Figure 4—figure supplement 1, Figure 4—figure supplement 2) comparing several LNK-based simulations to the DivS results. This hopefully further clarifies the relationships between these models.

The difficulty in fitting the LNK model is indeed a main limitation for its applicability more generally, and we have now made clear in the revised manuscript that the DivS model is a much more tractable alternative. For fitting the LNK model, while we are convinced that we have achieved optimal results for the examples considered here (following established methods), to our knowledge the LNK model will be limited to fitting relatively simple data. By comparison, the DivS model is relatively well-behaved (including requiring a fraction of the fitting time), and is much more flexible in describing both simulated data generated by more complex combinations of nonlinearities (Figure 4—figure supplement 2) as well as actual ganglion cell recordings (Figure 4). We thus have rewritten sections of the manuscript to highlight the flexibility and applicability of this modeling approach, rather than focusing on mechanistic differences (as in the previous manuscript).

*7) For the spiking DivS model, does the model really take the observed spike train as an input (not just as a constraint to fit data)? Shoudn't a model that takes the response as an input be able to produce the response? (It seems like cheating.) In fact, some studies that use the spike history in the way have shown that using the spike history is either the most important predictor of the response, or that the spike history alone can be used to predict the response (Kraus et al., 2015, Figure S4; Trucculo, et al., 2010).*

We are glad for this thoughtful concern, as it is potentially a subtle problem in previous studies using the spike history term, where this term is allowed to be positive. When it is positive, previous spikes can be used to predict the next spikes, and (as in the studies listed above), one can do quite well in using the spike-history term to predict future spikes, especially if spikes are correlated over long time scales. This is not as much of a concern in the retina, where time correlations are short (and there is much previous literature validating this approach in the retina, particularly from Paninski and Pillow).

However, we can eliminate this as a concern in this study for several reasons. First, as detailed in the methods, we constrain the spike history term to be negative, meaning it cannot be used to predict future spikes, but rather only influences temporal patterning on short time scales due to refractoriness. Second, as we demonstrate, using a spike history term with an LN model (the LN+RP model in later figures) has poor performance in high contrast (Figure 5), and also cannot explain contrast adaptation effects (Figure 7), and thus the spike-history term is only effective at explaining these effects in tandem with divisive suppression in a two-stage computation, which is a major point of this work.

*If this is really what was done, this approach differs from other models which use a spike-history term such as a generalized linear model (GLM) (Pillow et al., 2008) but that generate its own spikes rather than taking the actual response as an input. This needs to be made more clear, and also justified as to why this is useful. This approach may be useful for understanding the source of precision as the authors do, but it doesn't seem impressive that this model can fit the data if it really takes the actual response as an input.*

We hope that the clarification above explains how the model is more impressive given that it uses spike-history as a suppressive term that only has effects at a short time scale. Furthermore, the larger point of these figures on the spiking DivS model (Figure 6–Figure 8) is not that the spike-history term explains the spike response – quite the opposite. In these figures, we explicitly compare the spiking DivS model with the LN+RP model (without DivS) to show that DivS is necessary, and furthermore show that the DivS-RP model (without spike history) also cannot explain the response to high precision and contrast adaptation, demonstrating that it is the interplay of spike-history effects with the DivS computation that uniquely explain the ganglion cell response and its adaptation to contrast. As described above, the use of spike-history is well vetted by much published work in modeling the retina, but the DivS model goes well beyond this to show it alone is not sufficient to explain ganglion cell firing, and must be combined with computations present at the input to ganglion cells.

*8) Figure 6, it is not clear where the noise comes from in the model, as there is no noise source. Is it all from the spike-history term, i.e. the actual data? It's not clear how we can evaluate different models using this approach, because the noise is not appropriate for each model. A better approach would be to model noise sources and put them in each different model, like an LNP model.*

We apologize for omitting this from the previous version of the manuscript, and have added a sentence describing this detail in the Methods (lines 621-623). We used a non-homogeneous Poisson process to generate spikes from predicted firing rates. This is standard practice in most models of the retina and other visual neurons, and initial work comparing inhomogeneous Poisson processes to more realistic (but less tractable) integrate-and-fire processes (e.g., Pillow et al., 2005) has shown they yield equivalent models (see Paninski, Pillow, and Lewi chapter (2007): “Statistical models for neural encoding, decoding, and optimal stimulus design”). In other words, while there is still great debate on the actual noise sources underlying the lack of reproducibility of spike trains from trial-to-trial (see for example our recent paper McFarland et al., 2016), previous work particular to the retina has shown that our assumptions have no discernible effect on estimation of the model components.

*Reviewer #2:*

*[…]*

*Major comment:*

*My only concern is about novelty: as noticed by the authors, another model, the LNK by Ozuysal and Baccus, has already shown its ability to predict ganglion cell responses at different contrasts. The performance of the current model is better than the LNK, but not by much for full field stimuli. The only stimulus where there is a significant difference, if I understood well, was the one composed of a center spot and an annulus stimulating the surround. There the model performed better, but the difference is rather small (around 7%).*

To address this and the Reviewer 1’s comments, we have rewritten the manuscript to emphasize that the DivS model is a different – and in fact more general – explanation for retinal processing, such that divisive suppression appears to be a central component of ganglion cell computation (see Author Point E above). In Figure 4, the reason that the differences between models are smaller was because the LN model itself performs very well in the center-annulus stimulus, and one of the key points of this figure is the LNK model (i.e., adding synaptic depression), does no better than the LN model, meaning that this explanation itself cannot explain the results. In contrast, the DivS model outperforms both, showing that these added nonlinearities are present in the retinal computation. We hope that by making this more explicit in Figure 4 (as well as accompanying Figure 4—figure supplement 1 and Figure 4—figure supplement 2), and recasting the main points of our manuscript (including greatly elaborating the distinctions between the LNK and DivS model), that the novelty of the results (and the approach in general) is clear.

*The model is conceptually novel, but relatively similar to previous works by Sahani and colleagues.*

Sahani et al. have pioneered the multi-linear model, which is two linear terms that multiply one another. Such a model is conceptually distinct from the DivS model, which involves the multiplicative interactions of LN components – the important distinction being the nonlinearities, and the means to estimate them. Indeed, multiplication between model components is a well-worn idea (such as “gain terms”) and present in some form in many previous models, and the novelty of the Sahani model (e.g., Ahrens et al., 2008a; Williamson et al., 2016) was the statistical application of multiplicative linear terms. Such a model is not sufficient to explain the interactions we see (as the nonlinear suppression is critical), and the novelty of the model is to introduce such multiplicative interactions between nonlinear terms in a tractable context that can be fit to both intracellular and extracellular data. We agree that clarifying the novelty of our modeling approach in the context of previous work, and in addition to adding supplemental figures addressing STC and the LNK models (Figure 2—figure supplement 1, Figure 4—figure supplement 1 and Figure 4—figure supplement 2), we have added to the Discussion in this regard, most notably on lines 342-351):

“While divisive gain terms have been previously suggested in the retina – particularly in reference to early models of contrast adaptation (Mante et al., 2008; Meister and Berry, 1999; Shapley and Victor, 1978) – critical novel elements of the present DivS model include the ability to fit the nonlinearities of both LN terms by themselves, as well as their tractability in describing data at high time resolution. The presence of nonlinearities that are fit to data in the context of multiplicative interactions distinguishes this model from multi-linear models (two linear terms multiplying) (Ahrens et al., 2008a; Williamson et al., 2016), as well as more generalized LN models such as those associated with spike-triggered covariance (Fairhall et al., 2006; Samengo and Gollisch, 2013; Schwartz et al., 2006). Furthermore the model form allows for inclusion of spike-history terms as well as spiking nonlinearities, and can be tractably fit to both synaptic currents and spikes at high time resolution (~1 ms).”

*So I think a more thorough comparison between the LNK model and this one would be welcomed. With such a small difference, one would like to be sure that the authors gave the LNK model its best chance of success. Here the methods do not include a text about how this model was fitted. More generally, more details would be welcomed about how the two models were fitted in the case of the compound stimulus (centered spot +annulus).*

We apologize for this omission of methods, and further agree (also with Reviewer #1) that the manuscript would benefit from much more elaboration about the LNK model, and its relationship with the DivS model (see Author Point E above). We have also added a section in the Methods (lines 559-568), panels in Figure 4, and two supplemental figures (Figure 4—figure supplement 1 and Figure 4—figure supplement 2) offering further detail about the relationships between the models, and especially how the DivS model is an advance.

[Editors' note: the author responses to the re-review follow.]

[…]

*Reviewer #2:*

*The paper has improved from the previous version, but a number of issues remain that were raised in the first review. The main positive points are that it captures contrast adaptation in a particular cell type with a simple model, and that it raises a new potential mechanism for adaptation using one pathway modulating another, which is different from the accepted explanation of synaptic depression. It can be made suitable for publication, but a number of things must be done.*

We are grateful for the additional suggestions and hope to have addressed the remaining issues completely in the revision, as described below.

*1) Spike history. For the spiking DivS model, the authors use the raw data as an input to the model when predicting the data. This seemed to be the case in the last review, although it was baffling to me and surprised both reviewers. Insufficient justification has been given for this procedure, and I can't conceive of any justification for it. The arguments given by the authors for this procedure are not persuasive, and as stated in the last review, this seems like cheating to get the model to predict more accurately.*

There is apparently a misunderstanding that we did not clear up in the last round of review. Our model predictions of the spiking data do not use the observed spike trains themselves. This was described in the Methods section in the previous draft:

“Note that for validation of spike-based models, we simulated individual instances of spike trains using a non-homogeneous Poisson process, and the model predictions were based on many repeats for which we generated a PSTH.”

In other words, all model performance measures reported in previous versions (and the current version) of the manuscript used simulated spikes, and not actual recorded spikes, to generate model predictions. This includes all reported results about the spike-based models (including the GLM) in Figure 5, Figure 6 and Figure 7 (and the associated text).

We regret that we may have confused the issue in the last rebuttal by arguing that it is theoretically possible to use previously observed spikes to validate model performance. Clearly there is a debate in the field about this issue. In this rebuttal, we do not seek to resolve this issue, because in the manuscript we have generated spikes without taking into account the recorded spikes, consistent with the Reviewer’s recommendation.

As a result, our manuscript follows the following recommendation from the editor’s letter:

“Option 2: Run the model again, but using a spike history filter generated from synthetic spikes rather than using the raw data in the model prediction as was done in the most recent revision.”

We are sorry it was not clear that we had already done this in previous versions of the manuscript. To make this point absolutely clear in this revision, we have updated the descriptions of all the methods to accentuate this point. We have updated the specific labels in the relevant modeling diagram (Figure 5).

Thus, we will not argue further the validity of using past observed spikes to evaluate model performance (below), although look forward to spirited debate in the future to ultimately resolve this issue (outside of the context of this review process).

*Some arguments given by the authors as to why it is ok to use the data to predict the data:*

*“First, as detailed in the methods, we constrain the spike history term to be negative, meaning it cannot be used to predict future spikes, but rather only influences temporal patterning on short time scales due to refractoriness.”*

*If the spike history term is negative, thus causing silences, it helps the model predict future silences, which the same thing as changing the prediction of future spikes.*

*And Figure 5 shows the authors' claim that it cannot be used to predict future spikes isn't correct. With the spike history coming from the data, the beginning of bursts become more sharp, meaning the spike history is long enough for the data spikes in one burst to influence onset of spiking in the model's next burst. The decreased jitter in the statistics for the first spike in a burst confirms this.*

*Furthermore, the spike history goes out at least to 40 ms, and the full duration isn't shown or stated. The fact that it is small at 40 ms doesn't matter, because at long timescales the integration over multiple spikes will produce a large cumulative effect.*

“Second, as we demonstrate, using a spike history term with an LN model (the LN+RP model in later figures) has poor performance in high contrast (Figure 5), and also cannot explain contrast adaptation effects (Figure 7), and thus the spike-history term is only effective at explaining these effects in tandem with divisive suppression in a two-stage computation, which is a major point of this work. “

*This just means that this inappropriate use of the data in the model is not sufficient alone to predict the response, but it is necessary. It is nonetheless using the data to predict the data.*

“In these figures, we explicitly compare the spiking DivS model with the LN+RP model (without DivS) to show that DivS is necessary, and furthermore show that the DivS-RP model (without spike history) also cannot explain the response to high precision and contrast adaptation, demonstrating that it is the interplay of spike-history effects with the DivS computation that uniquely explain the ganglion cell response and its adaptation to contrast. As described above, the use of spike history is well vetted by much published work in modeling the retina, but the DivS model goes well beyond this to show it alone is not sufficient to explain ganglion cell firing, and must be combined with computations present at the input to ganglion cells. “

*The use of spike history in a feedback loop as part of the model is acceptable of course, but using the data spikes in the model's prediction is not. It is not well vetted to use this procedure to draw conclusions about the model performance, and as mentioned in the previous review references exist to show that it can greatly improve a model prediction (Kraus et al., 2015, Figure S4; Trucculo, et al., 2010), which shows why it is not an acceptable procedure.*

*The following statements and conclusions are currently unsupported, and need to be supported by a model using a spike history feedback filter, and not spike history from the raw data. It's fine to take these conclusions about millisecond precision and full adaptation in the spiking model out of the paper, the paper could stand without them.*

We hope that it is now clear that we did exactly what is suggested above for the spiking-model predictions: spikes were generated using a Poisson process, which was iterated forward in time. The spike history for this simulated spike train depended solely on these past simulated spikes. We apologize for any confusion regarding these methods, and hope that the issue has been clarified.

*Abstract. "The full model accurately predicted spike responses with unprecedented millisecond precision, and accurately described contrast adaption of the spike train."*

*Introduction. "Ganglion cell firing, further shaped by spike generation mechanisms, could be predicted to millisecond precision."*

As a result, we expect that it is not necessary to change these above sentences, as the simulated spike train did actually capture the true spike train to millisecond precision (without using the raw data).

*Figure 5–Figure 6 indicate that without the raw data (Div – RP model), predictions are no better than an LN model. Claims about precision should be removed unless a full spiking model with feedback spike history (without the raw data) is implemented.*

We have clarified Figure 5 to demonstrate that the model does not use the observed spike train to make model predictions, and changed the label from “Observed spike train” to “Spike history”. Figure 6 is meant to demonstrate the contribution of spike history to explaining the full spike response. Given that we have now clarified that all measures are based on simulated spikes (and do not use the observed spike trains in any way), we believe Figure 6 is appropriately stating the results as stands.

*Subsection “Contrast adaptation relies on both divisive suppression and spike refractoriness” "Therefore, the two nonlinear properties of retinal processing, contrast adaptation and temporal precision, are tightly related mechanistically and can be simultaneously explained by the divisive suppression model."*

*Contrast adaptation in spiking for the Div – RP model is improved over an LN model, so that claim can be kept, although full adaptation is not captured.*

We believe that the clarifications to the methods also make changing these above sentences unnecessary.

*2) Lack of slow contrast adaptation. There is currently insufficient evidence addressing the lack of slow contrast adaptation, and in fact the authors perform an analysis that would obscure evidence of slow adaptation. Unfortunately, the authors have resisted my request to show the raw data from a 20 s segment of the recording, which would show the change to both high and low contrast and the full 10 s recording at each contrast. It's ok if there is some slow adaptation, but they have to show whether it's there or not.*

We apologize for not initially taking the suggestion of the reviewer, and had meant to satisfy the request with what was shown in the previous Figure 1—figure supplement 1. We now show the requested panels in a modified version of this Figure, which is now divided into two supplemental figures associated with Figure 1.

Specifically, we now show a single 20-second segment of the stimulus and synaptic current for an example ON-Α cell (Figure 1—figure supplement 1). We then further analyze this by presenting the mean and standard deviation of the synaptic current within 1-sec windows for this example cell, and also show this quantity averaged over the recorded population (Figure 1—figure supplement 1). There is very little discernable drift in the average synaptic current relative to variation in current driven by the stimulus (Figure 1—figure supplement 1). For completeness, we also show the same analysis for the spikes (Figure 1—figure supplement 1).

Consistent with the reviewer’s statement above, we would like to stress that the purpose of this supplemental figure is not to argue whether ON-Α cells have slow contrast adaptation. Rather, it is an explanation for why our model — which has no slow adaptation mechanisms present – is able to fit this data with unprecedented accuracy. We hope this supplemental figure makes clear that the magnitude of this slow adaptation — if it exists at all — is very small relative to the stimulus-driven variations in current (i.e., compare changes in mean with the standard deviation of the current, shown by the error bars in Figure 1—figure supplement 1).

Furthermore, as Reviewer #1 also noted, we agree that our revision needed to make clear that the divisive suppression model can explain fast contrast adaptation, but does not address slow adaptation, which is largely absent in this experimental context. We have thus made this point explicit in the revised manuscript, both in response to Reviewer #1’s comment (see above), as well as adding additional Discussion.

*Slow adaptation is revealed only in the change in the average membrane potential (Manookin & Demb, 2006). Gain and temporal filtering do not adapt slowly (Baccus & Meister, 2003), but statistics for gain and temporal filtering are the focus of the supplemental figure. The only evidence that might have been in the figure is the vertical position of the nonlinearity, which would show the change in average membrane potential.*

We agree — we left out the most important aspect to resolve this issue in the original Figure 1—figure supplement 1, and have now corrected this omission. We believe that the updated supplemental figure now has exactly what was requested, including directly addressing the vertical position of the nonlinearity between the first and last 3 seconds of each trial (Figure 1—figure supplement 1), which is predicted by the DivS model (as shown in a new Figure 3).

Furthermore, these observations of little-to-no slow contrast adaptation are not inconsistent with Manookin and Demb (2006). Despite the different species and stimulation protocols, the one example of an ON-Α cell shown in Figure 5 (Manookin and Demb 2006) also exhibited a much smaller and shorter after-hyperpolarization relative to OFF-Α cells, which were the focus of that study.

*But surprisingly, in the methods it states, citing (Chander & Chichilnisky, 2001) "The resulting nonlinearities were then aligned by introducing a scaling factor for the x-axis and an offset for the y-axis." This y-axis offset would remove the evidence of slow adaptation. This offset was not used in (Chander & Chichilnisky, 2001) and should not be used here.*

Chander & Chichilnisky (2001) developed this analysis to introduce the gain into the filter of the LN model itself based on spiking data, which does not have much of an offset to account for (see Figure 1—figure supplement 1). However, as Figure 1—figure supplement 1 shows, the current data has a consistent offset in the y-axis of the nonlinearity. Also, subsequent work addressed the importance of distinguishing between gain and offset (see for example Lesica and Stanley, Network, 2006). As a result, we introduced an offset parameter when aligning the nonlinearities across contrast, so that we could accurately perform the scaling to match the slopes of the nonlinearities (and then scale the corresponding filters accordingly).

Nevertheless, we agree with the reviewer that the offset adjustment should not be overlooked. In fact, we had not paid much attention to it until creating the new version of Figure 1—figure supplement 1, because the DivS model requires no offset adjustment to fit the contrast changes. Thus, this line of questioning has revealed an extra element of the data we believe is worth reporting. First, we now explicitly demonstrate the offset in Figure 1—figure supplement 1: both in the raw and average traces, as well as the LN model fits to the currents. [The offset is now also included in the LN models shown in Figure 1.] Second, we now report these offsets in this supplemental figure. Finally, we now demonstrate that the DivS model predicts this offset difference between high and low contrast without any parameter adjustment (Figure 3).

*To be clear, three things must be done to address whether there is slow adaptation. 1) A full raw trace must be shown. 2) Nonlinearities must be shown without manipulation of the y-axis offset. 3) Because (Manookin & Demb, 2006) shows that most slow adaptation decays by 1.5 s, they should show the membrane potential averaged over trials and binned in increments of no larger than 1 s across the full 20 s.*

As described above, we have explicitly performed (1) and (3) in the revision, and for (2) have now demonstrated explicitly the offset shift in the LN model, which can be reproduced by the DivS model.

*3) LNK model. The authors are trying to claim that a model of synaptic depression can't reproduce the effect that there is greater suppression is in one spatial location than another. It is good that the authors have tried to compare the DivS model with different types of LNK models having two pathways, but in order to rule out a possibility (localized suppression from a depression model), they have to show that they have sufficiently tried to make that possibility work.*

*The authors resisted my previous suggestion in the first round of review to consider that bipolar cells have center-surround receptive fields and that more than one bipolar cell feeds into a ganglion cell. This implies that for two pathways, one from the center and one from an annulus, each tested pathway should have an input from both central and peripheral regions. The DivS models was allowed to have two pathways, each with a different weighting from the center and surround, but the LNK model does not have such a mixture for each pathway. A model should be tested where each of the two pathways of the LNK model have a weighting of center and surround. To be clear, one pathway should represent bipolar cells in the center, and should have a stronger weighting from the center than the annulus. The other pathway should represent bipolar cells whose receptive field center is under the annulus region, and should have a stronger weighting from the annulus than the central spot.*

In our original submission, we had argued that models with suppression generated by synaptic depression will have suppression with the same spatial footprint as excitation. The models considered in Figure 4—figure supplement 2 were meant to implement the reviewer’s previous suggestion, by exploring the range of reasonable models where one or multiple components of synaptic depression were embedded in a more complex nonlinear circuit. As explained below, we believe the additional models suggested by the reviewer above are already encompassed by these previous examples, but for completeness have also implemented the reviewer’s suggestion explicitly in this rebuttal. To clarify this for the reader, we have also added an explanation of this in the figure legend for Figure 4—figure supplement 2.

In fact, the models in Figure 4—figure supplement 2 encompass the extreme cases of the Reviewer’s suggestion, and as a result are expected to evince the strongest differences from models considered in Figure 4—figure supplement 1. Specifically, these models are all trying to test alternative situations where the ‘spatial footprint’ of suppression (as measured by the DivS model) could be different than that of excitation as a result of synaptic depression alone. Figure 4—figure supplement 2 considered the case where there were two completely separate processes of synaptic depression: one in center and one in surround. It seems that the model the Reviewer suggests is where the difference in spatial footprint between these two processes is less extreme, since one process (dominated by the center) will have some input from the surround, and vice versa. Such a model would have effects in between the models considered in Figure 4—figure supplement 2 (where the two processes have completely separate footprints) and where the two processes have the same footprints. However, this latter model reduces to the model in Figure 4—figure supplement 1, since in this case the two processes would generate the same output.

Because we tested both “extremes” of the model suggested by the Reviewer – and both yield a result where the spatial footprint of the suppression matches that of excitation – we would not expect intermediate models to behave differently. To test this (following the suggestion of the reviewer), we have generated the models to the specification of the reviewer, and demonstrate this in Figure 8:

Author response image 1.**DOI:**
http://dx.doi.org/10.7554/eLife.19460.015

Here, we parametrically vary the relative weight of the center/surround components, to effectively move between the two extremes already in the supplementals (i.e., Figure 4—figure supplement 1 and Figure 4—figure supplement 2). All panels here are analogous to those shown in Figure 4—figure supplement 2, although with the updated model (at left). As expected, the DivS model still reveals a suppressive term matching that of excitation (lower right), unlike the DivS model fit to data (Figure 4). Rather than include these additional modeling results as yet another supplemental figure and risk confusing the reader (versus the more motivated models already in Figure 4—figure supplement 1 and Figure 4—figure supplement 2), we instead have elected to reference these observations in the caption of Figure 4—figure supplement 2.

*This fits most with known circuitry, because bipolar cells have center-surround linear receptive fields. This test will probably work as the authors predict, but the current comparisons are not adequate for their claim.*

Although bipolar cells do have center-surround receptive fields and individual ganglion cells could receive inputs from multiple bipolar cells, bipolar cells that are in the surround generally do not provide excitatory input to ganglion cells. Nevertheless, we view these models as “extreme” in order to illustrate the logical argument that the footprint of suppression generated by synaptic depression will recapitulate the footprint of excitation, and thus consider them in Figure 4—figure supplement 2.

The previously published LNK model used a 4-state kinetic model, the current version of the paper now shows that a 3-state model was used here. This means that the simpler model chosen here can't reproduce all of the dynamics of the previous model. This difference needs to be clearly pointed out and justified in the main text. This issue also fits with showing whether that there is not slow adaptation, because it may be that the reason that the LNK model slightly underperforms the DivS model is that they chose a model with more simplified dynamics, yet the cell has both fast and slow dynamics.

As we have demonstrated in the new Figure 1—figure supplement 1, there is very little slow contrast adaptation in the present recordings, and as a result (1) the dynamics of the LNK model corresponding to the slow inactivation state are not useful; (2) the DivS model, which has no slow contrast adaptation, can accurately fit this data. After testing the LNK model with 4 components, we reduced it to 3-components in the text, and now make this clear in the Materials and methods section.

*4) Comparison to Spike Triggered Covariance. There is some confusion in the presentation about what STC analysis reveals, and the relationship to the DivS model.*

*“We added Figure 2—figure supplement 1 to fully describe an STC analysis and added modeling data to Figure 2. As explained >above, we more clearly demonstrate a number of key differences of the DivS >model over the STC model, most notably the presence of divisive suppression. >We now make clear the caveat that other cell types could require additional >components to capture important features of the response in the Discussion.”*

*The following is a summary of the relationship, and it is basically in agreement with what is stated in the legend of Figure 2—figure supplement 1: STC analysis produces orthogonal vectors that define the stimulus subspace that drives the cell's response. It does not speak to the subsequent nonlinear mapping of that subspace to the response. In this case, if there is a two-dimensional subspace that defines the response, the filters of the DivS models must occupy the same subspace two STC eigenvectors. Thus this subspace can be found by a standard STC analysis not requiring a optimization procedure.*

*For the choice of nonlinear mapping of the subspace to the response, either there can be an n-D nonlinearity (2-D in this case), which would be the optimal solution, or a more simplified nonlinearity can be found. This is what the DivS model does, to simplify the 2-D nonlinearity to a product of 1-D nonlinear functions.*

We agree with everything stated above. An important omitted element in the above (as we describe further below) is that directions of the filters themselves (within the subspace) are meaningful in the DivS model (and not orthogonal). The multiplicative form of the DivS model is what determines these filters, and allows the divisive form of interaction between them to be detected.

*The reason I requested this detailed comparison was simply to avoid the mistaken conclusion that the DivS model is really a new class of model. It finds a reduced dimensional subspace just as STC does, and in fact such a standard technique can be used. The DivS model then does provide a set of constraints to reduce the complexity of the subsequent nonlinearity, reducing a 2-D nonlinearity to 1-D functions.*

It seems that we agree on the facts, but not the semantics, as to what is defined as a “new class of model”. The STC model is an LN model, where the ‘N’ can be a multi-dimensional nonlinearity, and in practice usually cannot be fit. The DivS model for synaptic currents is an LNxLN model, and for spikes has an additional (LNxLN)LN form. These are by definition distinct mathematical forms, and in addition require different methods to estimate their parameters. Because the DivS model has different mathematical form with completely different means to estimate, it seems a reasonable bar to highlight this distinction from STC. Importantly, Figure 2—figure supplement 1 demonstrates that STC cannot reveal the underlying LNxLN model. It furthermore cannot take into account the spike history (which we show in later figures to be necessary to understand the spike response), and thus cannot replicate the form of the spiking DivS model either.

While we believe this classifies the DivS model as a different class of model than the STC model, we certainly agree they have some similarities (as well as differences), which are explored in Figure 2—figure supplement 1. In previous work (Butts et al., 2011; McFarland et al., 2013), we have perfromed additional analysis of STC-based solutions relative to other modeling forms, including its probabilistic cousin that we have been referring to as a GQM (generalized quadratic model; presented initially in Park and Pillow, NIPS, 2011). We routinely use such models to achieve independent estimates of the filters spanning the stimulus subspace, and its use in this manuscript is not intended to show any disrespect to STC. Rather, its presentation illustrates how the DivS model is distinct, and how it is necessary to reveal the main results presented in this manuscript: namely the divisive interactions underlying ganglion cell computation. We also use Figure 2—figure supplement 1 to demonstrate that STC correctly identifies the filter subspace (consistent with our previous work, but now applied to current recordings rather than spikes), and thus remains a useful analysis tool.

*Even though the Figure 2—figure supplement 1 legend basically agrees with these statements, there are several statements and analyses in the paper that indicate a confusion:*

*Subsection “The nonlinear computation underlying synaptic inputs to ganglion cells”. However, the 2-D mapping between STC filter output and the synaptic current differed substantially from the same mapping for the DivS model* (*Figure 2—figure supplement 1*).

*This isn't possible if things were done correctly (and it seems to conflict with the Figure 2—figure supplement 1 legend), the STC subspace should be the same as the DivS subspace, and the 2-D nonlinearity from the STC subspace should be the same as the mapping from subspace to response for the DivS model. I think this is just a problem of how something is being stated, but it should be corrected/clarified.*

This are glad for this careful description, and see where the misunderstanding is, and how to clarify. Specifically, we demonstrated in Figure 2—figure supplement 1 that the covariance-model (COV) subspace is the same as the DivS subspace, which is explicitly stated there (and in the reviewer’s comments above). However, the 2-D nonlinearities using the DivS filters (Figure 2) versus the COV filters (Figure 2—figure supplement 1) are not simply rotations of each other (as implied by the reviewers comments) because the DivS filters are not orthogonal. The specific DivS filters lead the DivS 2-D nonlinearity to be multiplicatively “separable”, whereas such a multiplicative interaction is not clear in the COV 2-D. In the revised manuscript, we have rewritten the section describing these differences, and in particular include this explanation explicitly within the section (The nonlinear computation underlying synaptic inputs to ganglion cells).

*There is one potential difference between a full-2D nonlinearity from an STC subspace and the DivS model, and that is that the full 2-D nonlinearity may overfit, and thus the DivS model may impose a regularization that allows a better model of a test data set. If this is true, this is an interesting point to make, and it seems to be more relevant with the spiking model, because the authors say that can't fit the 2-D nonlinearity for spiking data (they can of course, but it must not be accurate).*

We agree with this point: namely the many fewer parameters of the DivS model allows it to be tractably fit to much less data. There is enough data to fit the full 2-D models to currents (both based on the DivS and STC filters; Figure 2), and the model performance of each (Figure 2) is only slightly hurt by overfitting. There are in fact two barriers to fitting the STC to spiking data: one being the number of parameters, as the reviewer suggests. The second – which prevents STC from discovering the full computational structure of ganglion cell responses – is its inability to simultaneously fit spike-history terms. With enough spike data, the methods we present here might still be able to measure a 2-D nonlinearity, but such a nonlinearity would mix the divisive and spike-history effects together. Thus, the overarching point is that the STC model could not reveal the structure of computation we report in this paper.

*In the same section, Thus, despite the ability of covariance analysis to nearly match the DivS model in > terms of model performance (Figure 2), it could not uncover the divisive interaction between > excitation and suppression* (*Figure 2*).

*That's not the point of STC analysis, it only acts to find the relevant subspace. A second step would define the nonlinear mapping from subspace to response, either preserving the full 2-D nonlinearity, which would be the optimal solution, or simplifying the 2-D nonlinearity to a more biological combination of 1-D pathways.*

The purpose of this sentence was not to comment on STC analyses, but rather to stress the insight gained by using the DivS model, which discovers a much simpler structure than a full 2-D nonlinearity.

*It seems what the analysis of Figure 2 has done was to use the eigenvectors as filters for subsequent 1-D nonlinearities. That doesn't make any sense, the ideal 1-D nonlinearities have to lie in the 2-D subspace, but the 1-D nonlinearities don't have to be the eigenvectors, or even be orthogonal. This is not part of a standard STC analysis, and there is no reason to suspect that this would work.*

We used the full 2-D nonlinear mapping to estimate model performance, as described in the Materials and methods section.

We also realized that this section in the methods was separated from the descriptions of other models of synaptic currents (since it described models of spikes as well). We have moved this methods section to reside with the other model descriptions.

*The point of my request to state the relationship between the DivS model and STC analysis was not to pit STC analysis against the DivS model, because they should yield equivalent results as the stimulus subspace, and that's as far as STC analysis goes. It was so say that STC analysis will find the equivalent subspace as the DivS model, and that the DivS model provides a way to simplify the nonlinear mapping.*

We had previously added this section about STC as requested in the first round of review in order to demonstrate what is novel about the DivS model: in being able to identify an underlying divisive interaction in synaptic input. We realize the critical difference between the two methods in relation to their different 2-D nonlinearities did not previously come across clearly, and hope to fixed this in the revision (as described above). In doing so, we expect the reasons for including STC (in addition to responding to the requests following the first round of review) are clear.

*Discussion section, paragraph two. The presence of nonlinearities that are fit to data in the context of multiplicative >interactions distinguishes this model from multi-linear models (two linear terms multiplying) >(Ahrens et al., 2008a; Williamson et al., 2016), as well as more generalized LN models such as >those associated with spike- triggered covariance.*

*This isn't clear, STC only identifies the subspace, and then multiplicative interactions can be identified within that subspace.*

We hope that we have clarified above that the multiplicative interactions revealed by the DivS model are evident in the 2-D nonlinearity in the STC model, due to the non-orthogonality of the DivS filters (even though these filters share the same subspace as STC). There are no current methods to analyze 2-D nonlinearities other than low-rank approximations (which we perform here) – and in fact most applications of STC do not even go as far as estimating the 2-D nonlinearity. Whether or not methods could be developed to search for divisive interactions using STC, the overarching focus of the manuscript is that there are divisive interactions governing ganglion cell computation, and a means to discover them using the DivS model.

*5) Circuit mechanisms underlying divisive suppression.*

*It is appropriate to discuss the possibility that inhibition does play a role in adaptation, but the authors have to mention the previous literature that says that inhibition isn't needed for contrast adaptation (Brown & Masland, 2001; Manookin & Demb, 2006). This is in their favor to do so, because they point out that their model suggests an unexpected mechanism for contrast adaptation.*

We thank the reviewer for pointing this out. We agree, and have incorporated this point into the Discussion section.

*There is another point that the author's may wish to mention that supports their argument. As they mention, On-α cells are more linear with a high spontaneous firing rate. Depression models rely on there being a change in the average level of synapse activation with contrast. With a linear cell, the average input to the synapse doesn't change much with contrast, and thus a change in the level of depression may not occur. Thus, inhibition from a modulatory pathway may be needed to create contrast adaptation for a more linear cell.*

We agree, and have incorporated this point into the Discussion section.

*6) Meaning of Divisive Suppression. The authors use the term divisive suppression to apply both to one pathway modulating another (the usual term), and to any change in gain as might occur from synaptic depression. This second use is strange, if that's true, then the finding of divisive suppression isn't new, they have just called gain control something else.*

*The potential new concept is that the effect is better modeled by one pathway modulating the other. This is actually an old concept, (Victor) and has since been replaced by one where a second pathway is not necessary, namely synaptic depression. But the new result would be that the older concept of modulation works better, especially when one considers the spatial stimulus. To make this comparison and state this conclusion, it makes more sense to restrict 'divisive suppression' to mean modulation, and thus the question becomes one of Suppression vs. Depression, rather than saying depression is one type of suppression.*

We have not meant to claim that the idea of one pathway modulating another is a novel idea (even as a multiplicative/divisive interaction). This is the basis of most general models of gain control, as well as cortical models of “divisive normalization” (e.g., Carandini and Heeger, 2015). In fact, a divisive gain driven by presynaptic inhibition has been seen in the olfactory system of flies (Olsen and Wilson, 2008. Likewise, a canonical model of synaptic depression models it as a divisive gain (Markram et al., 1998). We include both possibilities in describing formulation of the DivS model (subsection “The nonlinear computation underlying synaptic inputs to ganglion cells”), and further address the relationship to other types of divisive suppression in the Discussion (subsection “Circuits and mechanisms underlying the divisive suppression”). The novelty of our work, in this respect, is revealing how such divisive suppression participates in shaping the ganglion cell response through computations within the retinal circuit.